# Species pool and soil properties in mangrove habitats influence the species-immigration process of diazotrophic communities across southern China

Nengjian Liao,[1,2,3] Lianghao Pan,[4] Huaxian Zhao,[5] Shu Yang,[5] Xinyi Qin,[5] Jiongqing Huang,[6] Xiaoli Li,[6] Ke Dong,[7] Xiaofang Shi,[4,8] Qinghua Hou,[1] Qingxiang Chen,[1] Pengbin Wang,[9] Gonglingxia Jiang,[1] Nan Li[1,2]

**ABSTRACT** Microbial immigration is an ecological process in natural environments; however, the ecological trade-off mechanisms that govern the balance between species extinction and migration are still lacking. In this study, we investigated the mechanisms underlying the migration of diazotrophic communities from soil to leaves across six natural mangrove habitats in southern China. The results showed that the diazotrophic alpha and beta diversity exhibited significant regional and locational variations. The diazotrophic species pool gradually increased from the leaves to nonrhizosphere soil at each site, exhibiting a vertical distribution pattern. Mantel test analyses suggested that climate factors, particularly mean annual temperature, significantly influenced the structure of the diazotrophic community. The diazotrophic community assembly was mainly governed by dispersal limitation in soil and root samples, whereas dispersal limitation and ecological drift were dominant in leaves. Partial least squares path modeling revealed that the species pool and soil properties, particularly the oxidation-reduction potential and pH, were closely linked to the species-immigration ratio of diazotrophic communities. Our study provides novel insights for understanding the ecological trait diversity patterns and spread pathways of functional microbial communities between below- and aboveground habitats in natural ecosystems.

**IMPORTANCE** Environmental selection plays key roles in microbial transmission. In this study, we have provided a comprehensive framework to elucidate the driving patterns of the ecological trade-offs in diazotrophic communities across large-scale mangrove habitats. Our research revealed that *Bradyrhizobium japonicum*, *Marinobacterium lutimaris*, and *Agrobacterium tumefaciens* were more abundant in root-associated soil than in leaves by internal and external pathways. The nonrhizospheric and rhizospheric soil samples harbored the most core amplicon sequence variants, indicating that these dominant diazotrophs could adapt to broader ecological niches. Correlation analysis indicated that the diversities of the diazotrophic community were regulated by biotic and abiotic factors. Furthermore, this study found a lower species immigration ratio in the soil than in the leaves. Both species pool and soil properties regulate the species-immigration mechanisms of the diazotrophic community. These results suggest that substantial species immigration is a widespread ecological process, leading to alterations in local community diversity across diverse host environments.

**KEYWORDS** mangrove ecosystems, diversity, species pool, driving factors, species-immigration mechanisms

Address correspondence to Nan Li, nli0417@163.com.

Nengjian Liao and Lianghao Pan contributed equally to this article. Author order was determined both alphabetically and in order of increasing seniority.

The authors declare no conflict of interest.

See the funding table on p. 18.

Microbial immigration or transmission is a widespread ecological process in natural environments (1–4), and environmental selection and dispersal play key roles in

shaping microbial communities (1, 3, 4). In forest environments, the plant microbiome is structured according to environmental conditions, plant traits, and climate change (5–8). Plant roots are inhabited by a complex inside (endosphere) or outside (rhizosphere) microbiota that is largely derived from the surrounding soil (6, 7). Root-associated microbiota can benefit the host plant by improving nutrient availability (i.e., phosphorus and nitrogen elements) (9, 10) and facilitating plant resistance to biotic and abiotic stresses by influencing plant hormone pathways (11). The microbes in endophytes and rhizospheres that are horizontally transferred through the soil also spread horizontally among the plant leaves and stems (8). Increasing evidence suggests that large populations of microorganisms inhabit leaves (12–15) and that these leaf endophytic microbes also have important effects on nutrient acquisition and plant health (15–17). To explore transmission processes at both the individual plant and ecosystem levels, we need to study endophyte colonization routes and dispersal modes; for example, microbial communities assemble and immigration mechanisms between belowground (i.e., soil and root) and aboveground (i.e., leaf) host environment. Natural ecosystems, especially mangrove ecosystems, are large nutrient stores that serve as reactors for biogeochemical cycles, providing excellent living conditions for certain functional microbes (e.g., carbon- and nitrogen-fixing communities) (18, 19). Mangrove ecosystems are rich in organic carbon and nitrogen, making them ideal habitats for microbial communities. It is necessary to evaluate the migration or transmission processes between below- and aboveground habitats in the functional microbiome.

Diazotrophic communities are essential components of the nitrogen cycle of terrestrial ecosystems, transforming substantial amounts of atmospheric $N_2$ into available N via biological nitrogen fixation (20, 21). Molecular methods based on universal PCR detection of *nifH* marker genes have been successfully applied to detect diazotroph communities in the natural environment (22–24). Compared to free-living N fixation diazotrophs, symbiotic diazotrophs may have an advantage because they live within plant tissues, where better niches are established for $N_2$ fixation and assimilation (25, 26). Recent evolutionary origins and ecological studies of nitrogen fixation (*nif*) genes and their genomic information (27, 28) indicate that nif-carrying free-living members derived from diverse soil samples have independently evolved from symbiotic ancestors. Furthermore, recent studies have shown that co-occurrence of endophytic diazotrophs is widespread in roots, stems, and leaves (29–31). These studies suggest that plant roots harbor diverse symbiotic diazotrophs that are genetically adapted to a dynamic environment. They can be naturally transferred through the soil, spread systemically, and reach aerial components. However, the transmission processes and environmental selection of rhizospheric and endophytic diazotrophs in natural ecosystems, particularly subtropical forest ecosystems, remain poorly understood.

Mangroves, as highly productive subtropical forest ecosystems, can be largely attributed to the high nitrogen-fixing activity of diazotrophs in mangrove sediments, which contribute approximately 40%–60% of the total nitrogen needed by mangrove ecosystems (32, 33). Mangrove ecosystems are considered typical nitrogen-limited ecosystems because of their tidal fluctuations and high denitrification rates (34). Therefore, as a key step in nitrogen cycling, nitrogen fixation is particularly important for providing available nitrogen to mangrove ecosystems. In recent years, high-through-put sequencing has offered a more comprehensive perspective on the roles of mangrove tree species and geochemical parameters in shaping diazotrophic communities in root-associated soil (35–39). For example, Zhang et al. (38) found that sulfate-reducing bacteria, including *Desulfobacteraceae* and *Desulfovibrionaceae,* were the dominant rhizosphere diazotrophic taxa in the Sanya River Mangrove Nature Reserve and that the dominant diazotrophs of the family *Desulfobacteraceae* were positively correlated with total phosphorus. Liu et al. (39) found harmonious coexistence patterns between rhizosphere and endosphere diazotrophs in the mangrove species *Avicennia marina* (Forsk.) on small and fine spatial scales, and iron (Fe) and pH were the major factors driving the divergence of endophyte-rhizophyte diazotrophs in mangrove ecosystems.

However, biogeographic studies on the migration of diazotrophic communities from below- to aboveground mangrove ecosystems at large scales have seldom been conducted.

Here, we collected sediment, rhizosphere soil, root, and leaf samples from the popular species *Kandelia obovata* from coastal mangroves across six natural coastal mangrove wetlands in China, spanning 1,150 km with a latitudinal range from 18.44°N to 24.39°N and a longitudinal range from 108.24°E to 117.92°E. We assessed the community structure and immigration process of diazotrophic communities using *nifH* gene amplicon sequencing, investigated shifts and assembly routes of diazotrophs, and identified the influencing factors and biogeographic changes in diazotrophic communities from soil to aerial tissues in mangrove forests. This work is crucial to facilitate a better understanding of the immigration mechanisms of diazotrophic communities across mangrove habitats.

## MATERIALS AND METHODS

### Study area and field sampling

The sampling site was located in coastal mangrove wetlands along a latitudinal gradient transect, spanning a wide geographic range (18.44°–24.39°N and 108.24°–117.92°E) (Fig. 1a). We collected 600 individual samples of *Kandelia obovata* from six accessible natural mangrove areas in southern China in July 2019. Five plots (5 × 5 m$^2$) were established for sampling in each region. These samples were fractionated into four components (Fig. 1b): nonrhizosphere soil (NS), rhizosphere soil (RS), root endosphere (RE), and leaves (LE) with 150 samples per type. These components were selected as models to elucidate the underlying immigration mechanisms of the diazotrophic communities from the soil to the leaves. The nonrhizosphere soil was shaken off the roots, whereas the rhizosphere soil (~1 mm thickness around the root) was washed with sterile water. The root samples were washed thrice to remove any remaining soil. The leaf litter of *K. obovata* was immediately washed with sterile water, freeze-dried, weighed, and ground. All samples were stored at −80°C until DNA was extracted.

### Biochemical factor analyses

Biochemical parameters for soils in the nonrhizosphere were measured (see supplemental Materials and Methods). We measured the oxidation-reduction potential (ORP), salinity, pH, and temperature. Soil texture (sand, silt, and clay) was determined using a Malvern Mastersizer 2000 (Malvern, United Kingdom). Total carbon (TC), total organic carbon (TOC), total nitrogen (TN), total sulfur (TS), total inorganic carbon (TIC), total phosphorus (TP), $SO_4^{2-}$, $PO_4^{3-}$, and inorganic nitrogen ($NO_2^-$-N, $NO_3^-$-N, and $NH_4^+$-N) were measured in the laboratory. Biotic factors (total plant coverage, *K. obovata* number, *K. obovata* average height, *K. obovata* basal diameter, *K. obovata* canopy area, and *K. obovata* coverage) were estimated in 5 × 5 m plots during the field survey. In addition, *K. obovata* aboveground biomass, *K. obovata* belowground biomass, and *K. obovata* total biomass were calculated using allometric equations by Rahman et al. (40). Data on climatic factors, including the mean annual temperature (MAT), mean annual precipitation (MAP), mean annual evaporation (MAE), mean annual relative humidity (MAR), and mean annual sunshine (MAS) duration, were collected from the China Meteorological Data Sharing Service System (http://data.cma.cn/) (Table S1).

### DNA extraction and PCR amplification

Approximately 0.25 g of frozen soil samples was used for DNA extraction using a Power Soil DNA Kit (MO BIO, Carlsbad, CA, USA). Approximately 0.05 g of root endosphere samples was extracted using a Power Plant Pro DNA Isolation Kit (MO BIO Laboratories, Inc., Carlsbad, CA, USA) after thorough grinding in liquid nitrogen. Leaf DNA was extracted using a GeneJET Genomic DNA Purification Kit (Thermo Scientific, Waltham,

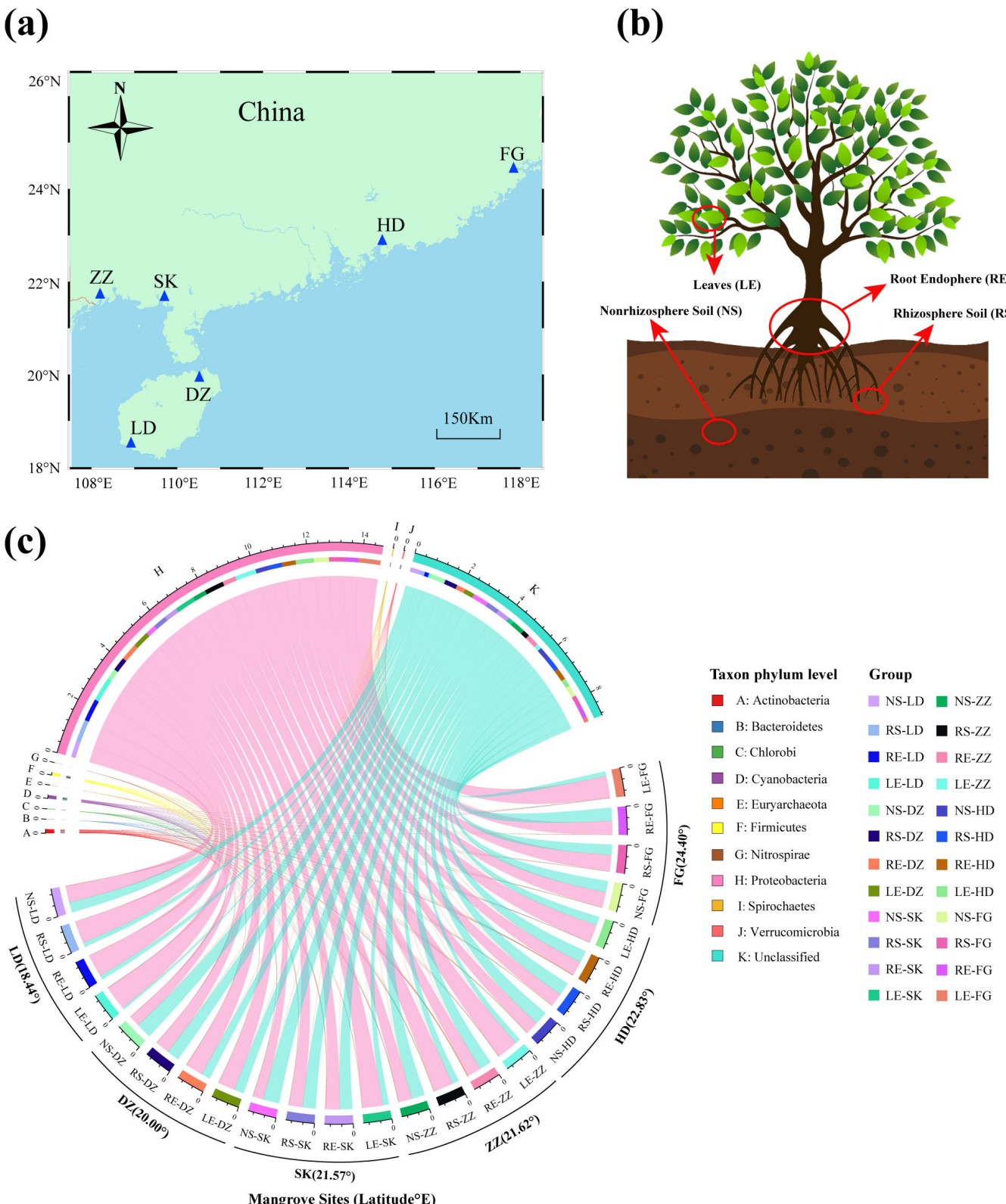

**FIG 1** Mangrove site location and community composition of diazotrophs in different sites and components. (a) Distribution of six sampling sites of mangrove sites in southern China. Two sites were from Hainan province (DZ and LD), two sites were from Guangxi province (ZZ and SK), one site was from Guangdong province (HD), and one site was from Fujian province (FG). (b) Schematic overview of nonrhizosphere sample, rhizosphere sample, root endosphere sample, and leaves sample. (c) Distribution patterns of the top 10 at phylum levels of the diazotrophic community are displayed using Circos analysis. LD, Ledong mangrove site; DZ, Dongzhaigang mangrove site; SK, Shankou mangrove site; ZZ, Zhenzhuwan mangrove site; HD, Huidong mangrove site; and FG, Fugong mangrove site.

USA) (41). The functional gene *nifH* was amplified using *nifH*-F and *nifH*-R primers (42) to investigate the diazotrophic community. High-throughput sequencing of the *nifH* gene was conducted on an Illumina MiSeq platform using a $2 \times 250$ bp Reagent Kit v2.

## Bioinformatics analysis

Raw sequences were analyzed to remove sequences with primer mismatches or lengths of <275 bp, low-quality reads (quality scores < 30), and barcode sequences. Chimera filtering and trimmomatic quality control were performed using the DADA2 denoising method in QIIME2 (43). High-quality sequences were combined and grouped into amplicon sequence variants (ASVs) without imposing an arbitrary dissimilarity threshold (44). Taxonomic assignment was based on both the nucleotide and translated amino acid sequences (44). The original sequence data obtained from our study were submitted to the National Center for Biotechnology Information Sequence Read Archive database under BioProject Accession PRJNA770021 and PRJNA1130613.

## Data analysis

The relative abundance-weighted community degree (RACD) was calculated to examine variations in diazotrophic community associations by different components as previously described (45). In the present study, we evaluated the immigration processes of the diazotrophic communities across different components of a large-scale mangrove habitat. Simm was the number of immigrating ASVs (e.g., in the immigration process from NS to RS, Simm indicates ASVs detected only in NS), Sext was the number of extinct ASVs (ASVs lost from the previous sample), and Stot was the total number of ASVs in both samples involved in the immigration process. The overall change in species composition was measured using the richness-based species exchange ratio (SERr), which was calculated as follows: SERr = (Simm + Sext)/Stot (46). The species extinction ratio (SExR) and species immigration ratio (SImR) were defined as Simm/Stot and Sext/Stot, respectively (45). The alpha-diversity (Shannon index) of each sample between different components was calculated by the "vegan" package using R software (version 4.2.0). Nonmetric multidimensional scaling (NMDS) and analysis of similarity (ANOSIM) were used to estimate the difference in community structure between different samples based on the Bray–Curtis dissimilarity using the "vegan" package and visualized using the "ggplot2" package (47). The species pool of the diazotrophic community was estimated using the "specpool" function in "vegan" package (48). Nonparametric testing (KwWlx) and Wilcoxon test were performed to reveal the differences using the "Easy-Stat" and "stats" packages, respectively (49). Mantel test analysis was used to further identify the driving factors explaining community variations and assembly mechanisms by the "vegan" package. The community assembly of the diazotrophic communities was evaluated as described previously by Li et al. (50). Variation partitioning analysis (VPA) was conducted to examine the contribution of biotic and abiotic factors to the community assembly according to the RDA (51). The linear regression analysis was performed using the "ggplot2" and "ggpmisc" packages. The shared ASVs and species between the different components were defined as Groups I–IV as previously described by Zhang et al. (52). Pearson's rank correlation between the immigration mechanisms of the diazotrophic community and environmental factors was calculated using the "psych" package. Partial least squares path modeling (PLS-PM) was conducted to determine the direct and indirect contributions of driving factors on the SImR by the "plspm" package.

Further details and other statistical analyses are provided in the supplemental Materials and Methods.

## RESULTS

### Community composition and diversity dissimilarity within mangrove habitats

After quality filtering and barcode sequence removal, 9,384 ASVs were detected using high-throughput sequencing of the *nifH* gene. Diazotrophic communities were dominated by ASVs belonging to the phyla Proteobacteria (61.18%), Actinobacteria (0.58%), Cyanobacteria (0.48%), Firmicutes (0.38%), Verrucomicrobia (0.18%), Spirochaetes (0.12%), and unclassified diazotrophs (37.01%) (Fig. 1c). Proteobacteria was the most abundant phylum in all samples, accounting for 90.04% of the LE-LD-18.44° group (Fig. 1c), and the most dominant diazotroph was *Bradyrhizobium* sp. *IRBG 228* (Fig. S1, 15.61%). At the species level (Fig. S1), these dominant diazotrophs were *Bradyrhizobium japonicum* (5.15%), *Marinobacterium lutimaris* (4.48%), and *Agrobacterium tumefaciens* (2.85%).

Alpha diversity (Shannon index) was found to gradually increase from leaves to nonrhizosphere soil, and significant differences ($P < 0.05$) were observed between different components using nonparametric testing (Fig. 2a). The highest and lowest alpha diversities were found in the NS-ZZ-21.62° and LE-LD-18.44° groups, respectively (Fig. 2b). Furthermore, there were significant differences ($P < 0.05$) between the different components of each mangrove habitat (Fig. 2b). The results revealed that alpha diversity was higher in the soil than in the leaves. Similarly, the species pools of the nonrhizosphere and rhizosphere soils were significantly larger ($P < 0.05$) in the diazotrophic

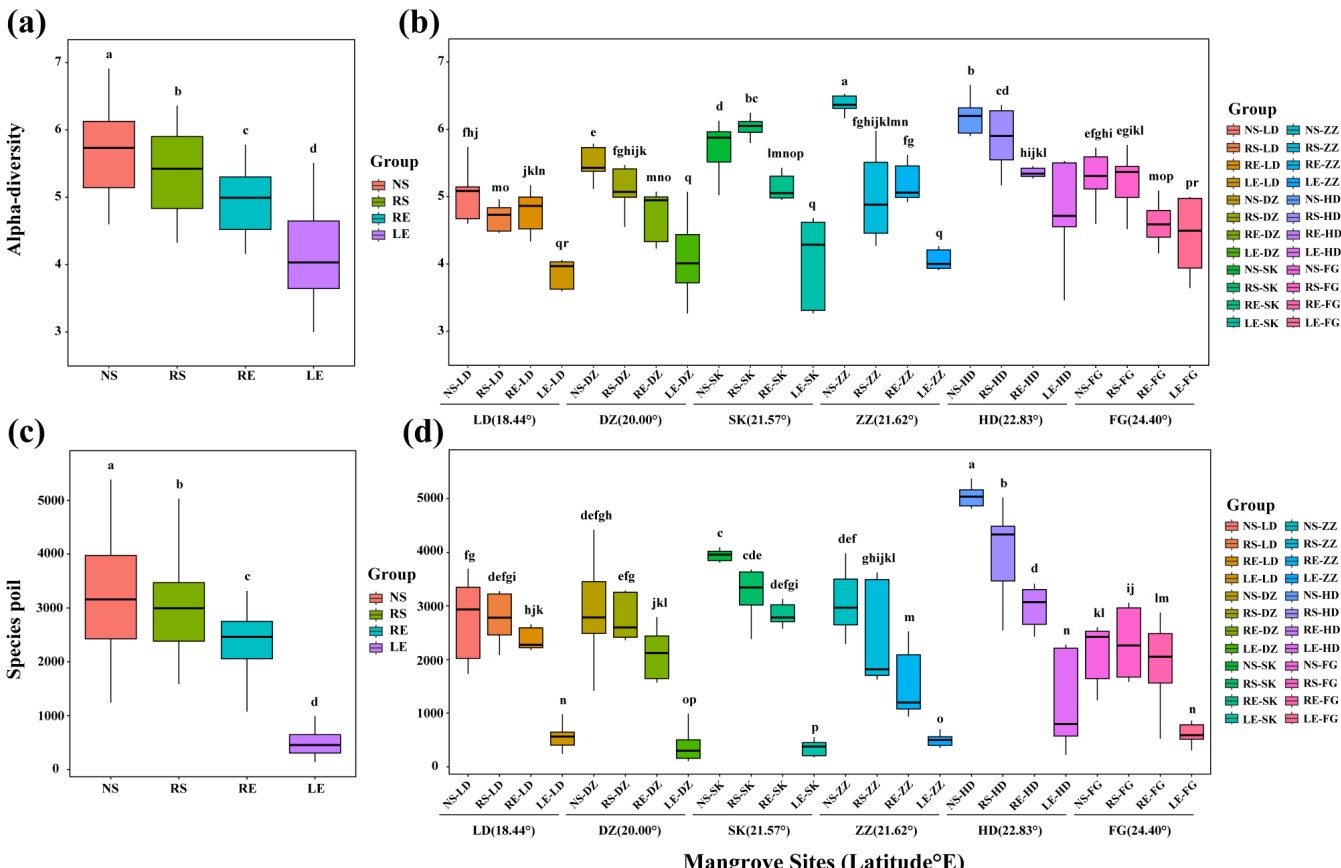

**FIG 2** Variation of diazotrophic communities in different groups. The alpha diversity (Shannon index) (a) and species pool (c) across different components are presented using boxplots. The alpha diversity (b) and species pool (d) in different sites and components along the latitudinal gradient are presented by boxplots. Different letters above the box plot indicated a significant difference ($P < 0.05$) using non-parametric testing (KwWlx). LD, Ledong mangrove site; DZ, Dongzhaigang mangrove site; SK, Shankou mangrove site; ZZ, Zhenzhuwan mangrove site; HD, Huidong mangrove site; and FG, Fugong mangrove site.

communities than in the leaves (Fig. 2c). The species pool of the diazotrophic community of leaves in each habitat, except HD-22.83°, was smaller, whereas a larger species pool was observed in the nonrhizosphere and rhizosphere soil groups in the HD-22.83° group (Fig. 2d). The diazotrophic species pool exhibited a vertical distribution pattern extending from the leaves to the nonrhizosphere soil (Fig. 2c and d), wherein the diazotrophic species pool peaked in the nonrhizosphere soil and gradually decreased toward the leaves. Based on Bray–Curtis distance (beta-diversity), NMDS of the diazotrophic community revealed significant differences between diazotrophic communities and distinct clusters in different components (ANOSIM, $R = 0.33$–$0.88$, $P < 0.001$) (Fig. S2). In contrast, a greater β-deviation of the diazotrophic community was observed in the nonrhizosphere and rhizosphere soil samples (Fig. S2a and b). These results indicated that the soil environment drove convergence in the diazotrophic community composition. Furthermore, the relationship between geographic distance and biodiversity (i.e., alpha diversity, beta diversity, and species pool) was constructed using linear regression models (Fig. S3a through c). The results showed that diazotrophic biodiversity significantly increased ($P < 0.001$) with increasing geographical distance in all groups. The species pool of the NS group had a steeper slope than those of the other groups (Fig. S3c), indicating that the species pool in the soil had a greater change than that of the endophyte-rhodophytes across large geographic scales. In addition, linear regression analysis showed a positive correlation ($P < 0.001$) between species pool and diversity (alpha diversity and beta diversity) (Fig. S4), indicating resource availability, interspecific relationships, and habitat differences have significant influences on species pool (53).

## Diazotrophic community structure driving factors and assembly processes

Mantel test analysis was used to further identify the potential biotic and abiotic factors that explain the variations in the diazotrophic community (Fig. 3). We found that MAT, salinity, plant total coverage (PTC), and TOC were significantly and positively correlated with the diazotrophic community across all groups, with MAT ($R = 0.318$–$0.432$, $P < 0.001$) emerging as the most crucial driving factor. Soil texture, including sand, silt, and clay ($R = 0.217$–$0.426$, $P < 0.001$), was more important for the root-associated diazotrophic community than for the leaves. However, $SO_4^{2-}$ ($R = 0.322$, $P < 0.01$) was the main driving factor affecting the diazotrophic community in the rhizosphere soil. In the LE group, more factors exhibited either no correlation or a weaker positive correlation with the diazotrophic community. The above results show that both biotic and abiotic factors drive shifts in diazotrophic community structure across large-scale mangrove habitats.

In addition, null model theory was used to investigate the underlying effects of deterministic or stochastic processes on diazotrophic community structure. We found that the diazotrophic communities in the NS, RS, and RE groups were mainly shaped by dispersal limitations (Fig. S5a through c; NS: 63.79%; RS: 59.16%; and RE: 64.10%), followed by heterogeneous selection. Nevertheless, both ecological drift (35.95%) and dispersal limitation (26.31%) dominated the diazotrophic community assembly in the LE group (Fig. S5d). The results showed that stochastic processes are more important in determining the diazotrophic community assembly of different components across mangrove ecosystems. Next, we explored the correlations between driving factors and βNTI using Mantel tests (Table S2). The results showed that pH ($R = -0.044$–$0.1243$, $P < 0.001$) and TS ($R = -0.0728$–$0.0806$, $P < 0.001$) determined the changes in diazotrophic community structure, whereas ORP ($R = 0.1074$, $P < 0.001$) was significantly associated with nonrhizosphere community structure. To classify the importance of the biotic and abiotic factors in driving the assembly of diazotrophic communities, VPA was performed (Fig. S6). The VPA results showed that abiotic factors are more important for driving the assembly of diazotrophic communities, especially in the nonrhizosphere sample group. However, the assembly of diazotrophic communities was unexplained (84%–90%) regardless of different components (Fig. S6).

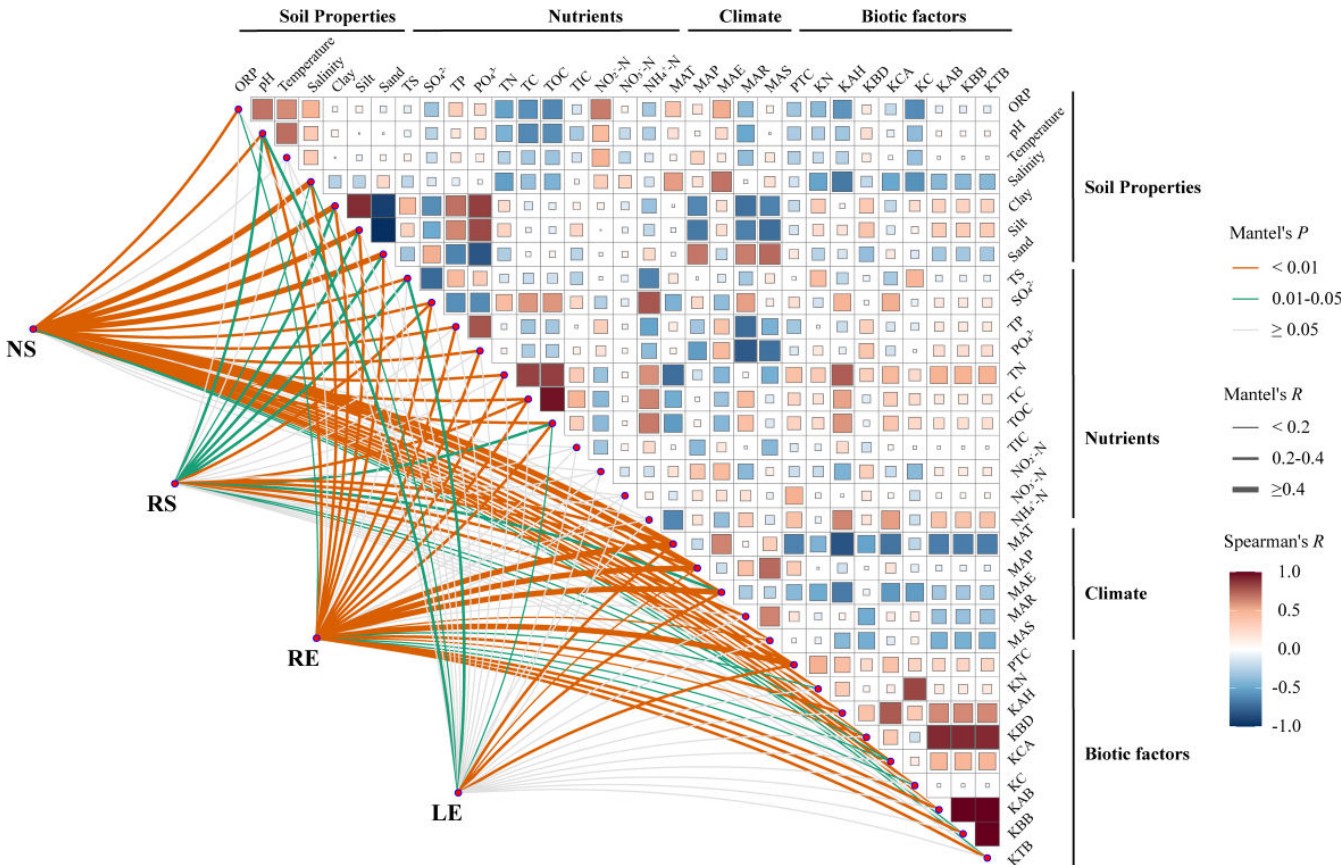

**FIG 3** Pair-wise correlations between environmental factors (top right) and different components based on Mantel tests. For the Mantel test results, the line color indicated the $P$ values, and the line size indicated the absolute value of Mantel's $r$. ORP, pH, temperature, salinity, and soil texture (clay, silt, and sand) were summarized as soil properties. TS, $SO_4^{2-}$, TP, $PO_4^{3-}$, TN, TC, TOC, TIC, $NO_2^--N$, $NO_3^--N$, and $NH_4^+-N$ were summarized as nutrients. MAT, MAP, MAE, MAR, and MAS were summarized as climate factors. PTC, KN, KAH, KBD, KCA, KC, KBB, KAB, and KTB were summarized as biotic factors. KN, *Kandelia obovata* numbers; KAH, *Kandelia obovata* average height; KBD, *Kandelia obovata* basal diameter; KCA, *Kandelia obovata* canopy area; KC, *Kandelia obovata* coverage; KAB, *Kandelia obovata* aboveground biomass; KBB, *Kandelia obovata* belowground biomass; and KTB, *Kandelia obovata* total biomass.

## Identifying shared ASVs and divergence in immigrant taxa across large-scale mangrove habitats

Linear regression analysis showed a positive correlation ($P < 0.001$) between geographic distance and RACD for the different components (Fig. S7a). The highest and lowest RACD were observed in the RE-LD-18.44° and LE-ZZ-21.62° groups, respectively (Fig. S7b). Furthermore, the RACD was generally higher in the soil and root-associated samples than in the leaves, implying closer associations between ASVs (Fig. S7b). The UpSet plots showed that unique ASVs were the highest in the nonrhizosphere soil, followed by rhizosphere soil (650 ± 161 and 435 ± 162, respectively), while root endosphere and leaves—structures that were above the ground line—had the lowest unshared ASVs, with 405 ± 91 and 160 ± 97, respectively (Fig. 4a). There were significant differences in the regional species pools, wherein the highest ASVs were shared between the different components in HD-22.83° and LD-18.44° with 2,202 and 1,137, respectively (Fig. S8). ZZ-21.62° and SK-21.57° exhibited the lowest number of shared ASVs (862 and 647, respectively). These results revealed that unique ASVs gradually decreased from the nonrhizosphere soil to the leaves (Fig. 4a; Fig. S8), indicating that the soil environment had a broader ecological niche.

To further predict the transmission of diazotrophs from below to aboveground, we attempted to identify two potential pathways in the soil−plant system at the ASV (Fig. 4b) and species levels (Fig. 4c). The transmission of ASVs from nonrhizosphere soil to

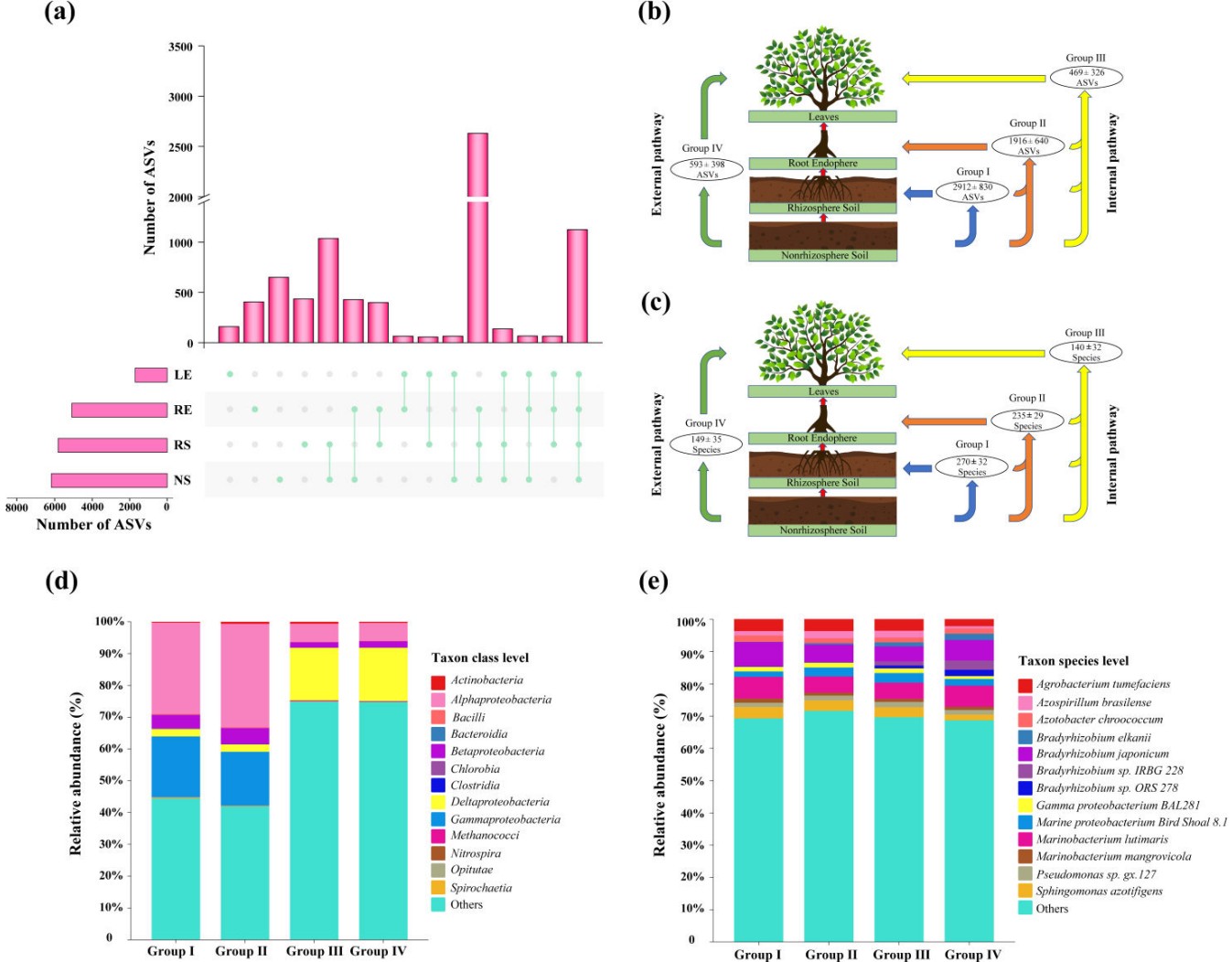

**FIG 4** Distribution patterns of the shared and unique ASVs in different groups. The UpSet plot was chosen to visualize the common and unique mean ASVs (a) across different components. The two potential immigration pathways of diazotrophic ASVs (b) and species (c) in different components from belowground (i.e., nonrhizosphere soil and rhizosphere soil) to aboveground (i.e., root endosphere and leaves) habitats were predicted. The shared ASVs and species from distinct components were defined as Groups I–IV (Group I: the ASVs shared between NS and RS; Group II: the ASVs shared between NS, RS, and RE; Group III: the ASVs shared between NS, RS, RE, and LE; and Group IV: the ASVs shared between NS and LE). Groups I, II, and III contained the shared ASVs (b) and species (c) that likely immigrated through the internal pathway; Group IV contained the shared ASVs (b) and species (c) that likely immigrated through the external pathway. Distributions of diazotrophic composition (top 13) in Groups I– IV at the class level (d) and species level (e), in which Group I indicated the ASVs shared between NS and RS; Group II indicated the ASVs shared between NS, RS, and RE; Group III indicated the ASVs shared between NS, RS, RE, and LE; Group IV indicated the ASVs shared between NS and LE.

leaves is defined as the internal pathway (52). The results showed that there were 2,912 ± 830 ASVs shared between nonrhizosphere soil and rhizosphere soil (Group I in Fig. 4b), 1,916 ± 640 ASVs were shared among nonrhizosphere soil, rhizosphere soil, and root endosphere (Group II in Fig. 4b), and 469 ± 326 ASVs were shared among four different components (Group III in Fig. 4b), indicating that these ASVs might immigrate from below- to aboveground through the interior of plant tissues. The transmission of diazotrophs by molecular motion and the diffusion of aerosols to the surface of leaves was defined as the external pathway (52). There were 593 ± 398 ASVs shared between nonrhizosphere soil and leaves in the external pathway (Group IV in Fig. 4b). This result indicates that the dispersal of microorganisms by external pathways occurred more frequently than transfer from roots to leaves via the internal pathway. Similarly,

we found that the shared species between different components showed a gradually decreasing trend from Group I to Group III (Fig. 4c), and the most shared species between nonrhizosphere soil and rhizosphere soil were 270 ± 32 in Group I. In addition, the shared ASVs and species at each site were differentially distributed along latitudinal gradients within the large-scale mangrove soil-plant system, wherein the highest shared ASVs and species, especially in Groups I and II, were observed in HD-22.83° (Fig. S9a and b). These results indicated that the soil environment has a greater prevalence of shared ASVs and species than other components, which may serve as a seed pool for root and leaf microbiome assembly via internal and external pathways. Furthermore, we analyzed the composition of immigrant diazotrophic taxa at the class level (Fig. 4d). Alphaproteobacteria, Betaproteobacteria, Gammaproteobacteria, and other diazotrophic taxa were more abundant in Groups I and II. However, Gammaproteobacteria were detected in extremely few taxa, whereas other supergroups were more abundant in Groups III and IV (Fig. 4d). In our study, the dominant diazotrophs in Groups I, II, and III were *Bradyrhizobium japonicum*, *Marinobacterium lutimaris*, and *Agrobacterium tumefaciens* (Fig. 4e). Similarly, we found *Marinobacterium lutimaris*, *Bradyrhizobium japonicum*, and *Bradyrhizobium* sp. *IRBG 228* in Group IV prevailed as the prominent diazotroph.

## Effect on the immigration process of diazotrophs by biotic and abiotic factors across habitats

In the present study, we observed an increase in the species exchange ratio and species immigration ratio of diazotrophic communities through immigration from the soil to leaves (Fig. 5). Conversely, the species-extinction ratio of diazotrophic communities decreased from the soil to the leaves. For large-scale mangrove habitats, the highest SERr and SImR of diazotrophic communities were found in the ZZ-21.62° group via the transmission pathway from nonrhizosphere soil to rhizosphere soil (NS→RS), as well as from rhizosphere soil to root endosphere (RS→RE) (Fig. S10.1, 3, 4, and 6). The lowest SExR of diazotrophic communities was found in the HD-22.83° group via the transmission pathway from nonrhizosphere soil to rhizosphere soil (Fig. S10.2). In addition, we found that the variation rates of community dynamics (SERr, SExR, and SImR) were generally higher in the group via the transmission pathway from the root endosphere to leaves (RE→LE) than in the other groups (Fig. S10.7 through 9). Nevertheless, we did not find any correlation between the variation rates of community dynamics and latitudinal gradients using the Wilcoxon test, indicating that species immigration mechanisms are widespread in large-scale mangrove habitat ecosystems. Linear regression analysis found that there were extremely positive correlations ($P < 0.001$) between immigration mechanisms (SERr, SExR, and SImR) and geographic distance in the groups via the transmission pathway from rhizosphere soil to root endosphere (RS→RE) (Fig. S11a through c). However, no significant correlation between SImR and geographic distance was observed only in the group via the transmission pathway from nonrhizosphere soil to rhizosphere soil (NS→RS) (Fig. S11a). Furthermore, SERr and SImR showed extremely negative correlations with alpha diversity in the groups via the transmission pathway from nonrhizosphere soil to rhizosphere soil (NS→RS) and from root endosphere to leaf groups (RE→LE) (Fig. S12a and g). In contrast, SExR and alpha diversity showed weaker positive correlations in the group via the transmission pathway from the root endosphere to leaves (RE→LE) (Fig. S12d). SERr, SExR, and SImR exhibited extremely positive correlations ($P < 0.001$) with beta diversity in all the groups (Fig. S12b, e, and h), and beta diversity may play a decisive role in shaping microbiome shifts. In addition, SERr and SImR exhibited strong negative relationships with the species pool in all groups compared to beta diversity (Fig. S12c and i), whereas they exhibited strong positive relationships between SExR and the species pool in the groups via the transmission pathway from rhizosphere soil to root endosphere (RS→RE) and from root endosphere to leaves (RE→LE) (Fig. S12f).

In addition, we estimated the effects on diazotrophic immigration of different components by biotic and abiotic factors across large-scale mangrove habitats based on Pearson's correlation analysis (Fig. 6). The results showed that TC, TOC, and MAR

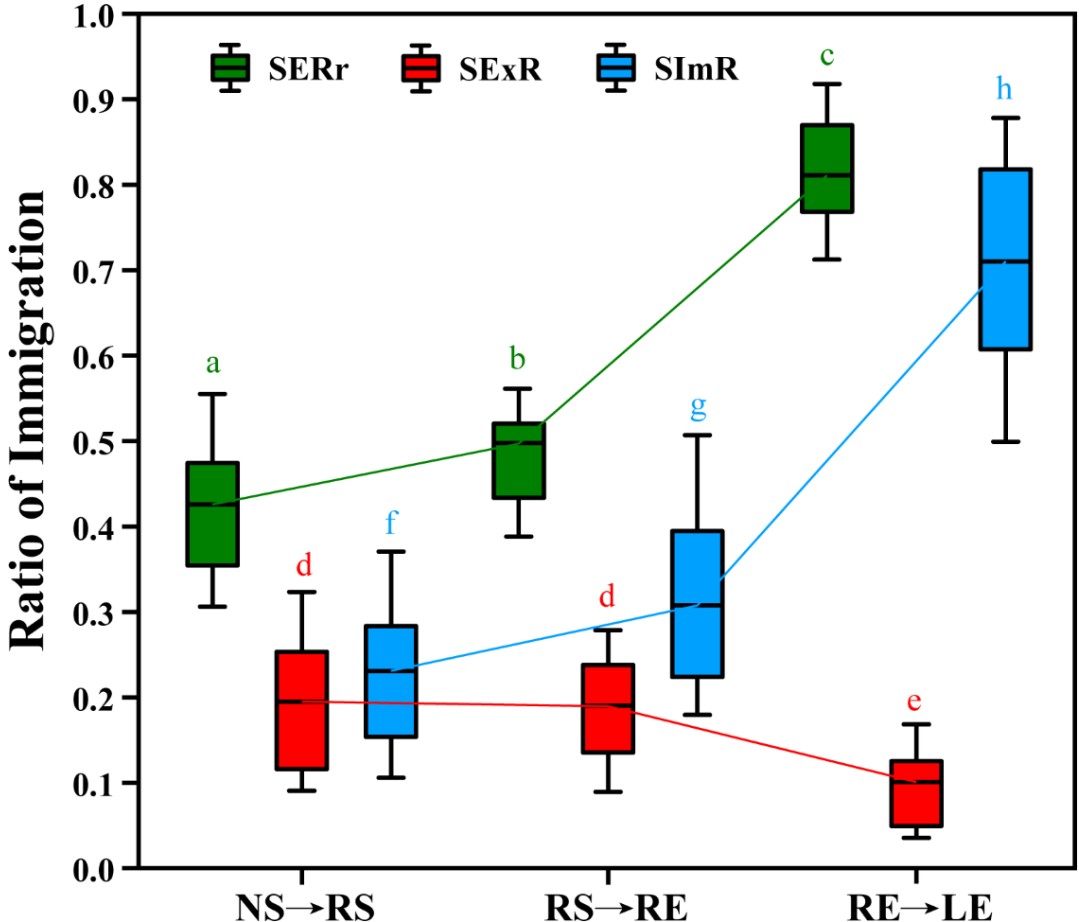

**FIG 5** Variations of the immigration process of diazotrophic communities (SERr, SExR, and SImR) in different components across mangrove habitats. Different letters above the box plot indicate a significant difference ($P < 0.05$) using the Wilcoxon test. NS→RS, the transmission pathway from nonrhizosphere soil to rhizosphere soil; RS→RE, the transmission pathway from rhizosphere soil to root endosphere; RE→LE, the transmission pathway from root endosphere to leaves.

were significantly positively correlated ($P < 0.05$) with SERr, and MAR also showed a significantly positive correlation ($P < 0.01$) with SImR in the groups via the transmission pathway from nonrhizosphere soil to rhizosphere soil (NS→RS) and from rhizosphere soil to root endosphere (RS→RE) (Fig. 6a and b). In contrast, ORP, temperature, and $NO_2^-$-N had a significantly negative correlation ($P < 0.05$) with SERr (Fig. 6a and b). However, we observed that more biotic factors (plant traits and biomass) had significantly positive correlations ($P < 0.05$) with SExR in the group via the transmission pathway from root endosphere to leaves (RE→LE) (Fig. 6c). In general, the factors significantly related to the immigration process of diazotrophic communities exhibited a gradual decreasing trend in the soil-plant system (Fig. 6a through c).

In this study, we focused on assessing the underlying influence of biotic and abiotic factors on the SImR. Considering the effects of environmental filtering, PLS-PM showed that alpha diversity (path coefficients = −0.105, $P < 0.01$), beta diversity (path coefficients = 0.45, $P < 0.001$), and species pool (path coefficients = 0.212, $P < 0.001$) jointly affected the SImR in the group via the transmission pathway from nonrhizosphere soil to rhizosphere soil (NS→RS), of which beta diversity and species pool showed direct and total positive effects, while alpha diversity exhibited direct and total negative effects (Fig. 7a). In addition, soil properties, geographic distance, and nutrients had a stronger negative direct effect (path coefficients: −0.111,−0.214, and −0.232, respectively; all $P < 0.001$) on the SImR, yet climate factors had a stronger positive direct effect (path coefficients: 0.419; $P < 0.001$), wherein climate factors showed greater total positive

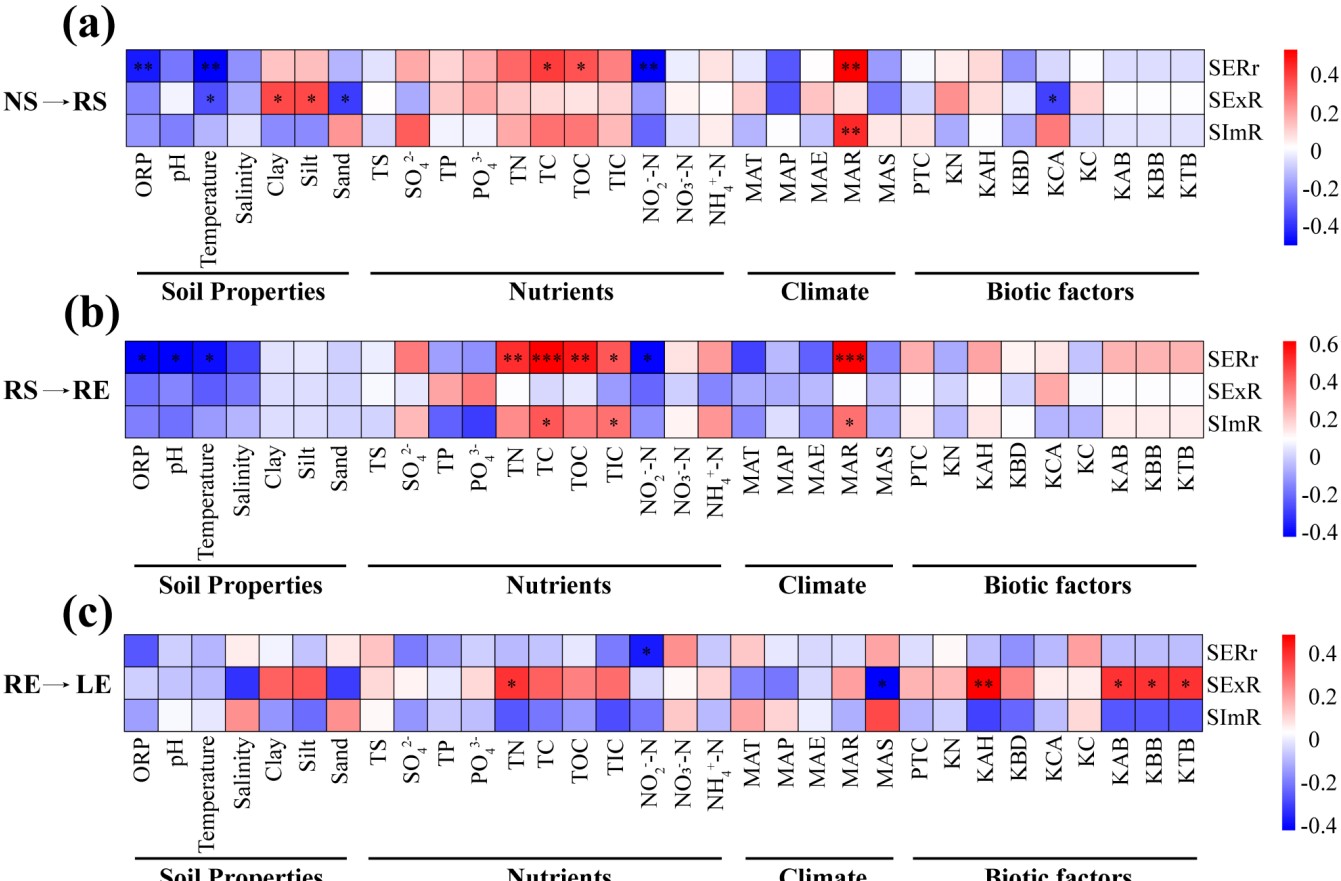

**FIG 6** A correlation heat map illustrating pair-wise relationships between SERr, SExR, and SImR and soil properties, nutrients, climate, and biotic factors across different components. (a) NS→RS, the transmission pathway from nonrhizosphere soil to rhizosphere soil; (b) RS→RE, the transmission pathway from rhizosphere soil to root endosphere; (c) RE→LE, the transmission pathway from root endosphere to leaves based on Pearson correlation coefficient analysis. Significance levels are indicated: $*P < 0.05$, $**P < 0.01$, and $***P < 0.001$. ORP, pH, temperature, salinity, and soil texture (clay, silt, and sand) were summarized as soil properties. TS, $SO_4^{2-}$, TP, $PO_4^{3-}$, TN, TC, TOC, TIC, $NO_2^--N$, $NO_3^--N$, and $NH_4^+-N$ were summarized as nutrients. MAT, MAP, MAE, MAR, and MAS were summarized as climate factors. PTC, KN, KAH, KBD, KCA, KC, KBB, KAB, and KTB were summarized as biotic factors. KN, *Kandelia obovata* numbers; KAH, *Kandelia obovata* average height; KBD, *Kandelia obovata* basal diameter; KCA, *Kandelia obovata* canopy area; KC, *Kandelia obovata* coverage; KAB, *Kandelia obovata* aboveground biomass; KBB, *Kandelia obovata* belowground biomass; and KTB, *Kandelia obovata* total biomass.

effects on the SImR (Fig. 7a). In the group via the transmission pathway from rhizosphere soil to root endosphere (RS→RE) (Fig. 7b), we found that alpha diversity (path coefficients = −0.144, $P < 0.01$), beta diversity (path coefficients = 0.133, $P < 0.001$), and species pool (path coefficients = 0.614, $P < 0.001$) were also significantly related to the SImR, of which beta diversity and species pool showed direct and total positive effects, while alpha diversity showed direct and total negative effects. Soil properties (path coefficient = −0.082, $P < 0.01$) and biotic factors (path coefficient = −0.075, $P < 0.05$) were significantly negatively correlated with SImR (Fig. 7b). In the group via the transmission pathway from the root endosphere to leaves (RE→LE) (Fig. 7c), we observed that beta diversity (path coefficients = 0.19, $P < 0.001$) and species pool (path coefficients = 0.577, $P < 0.001$) exhibited significantly positive direct and total effects on SImR. Additionally, only soil properties (ORP and pH) exerted a stronger negative effect (Fig. 7c). All the reflective indicators that were identified, including $NH_4^+-N$, ORP, pH, MAR, and KCA, exhibited factor loadings exceeding 0.6 on their respective latent variables within the modules in all groups (Fig. 7a through c). From the above results, we found that the species pool emerged as the most influential factor with a more positive effect on the SImR than beta diversity, whereas soil properties (ORP and pH) were identified as the most crucial factor

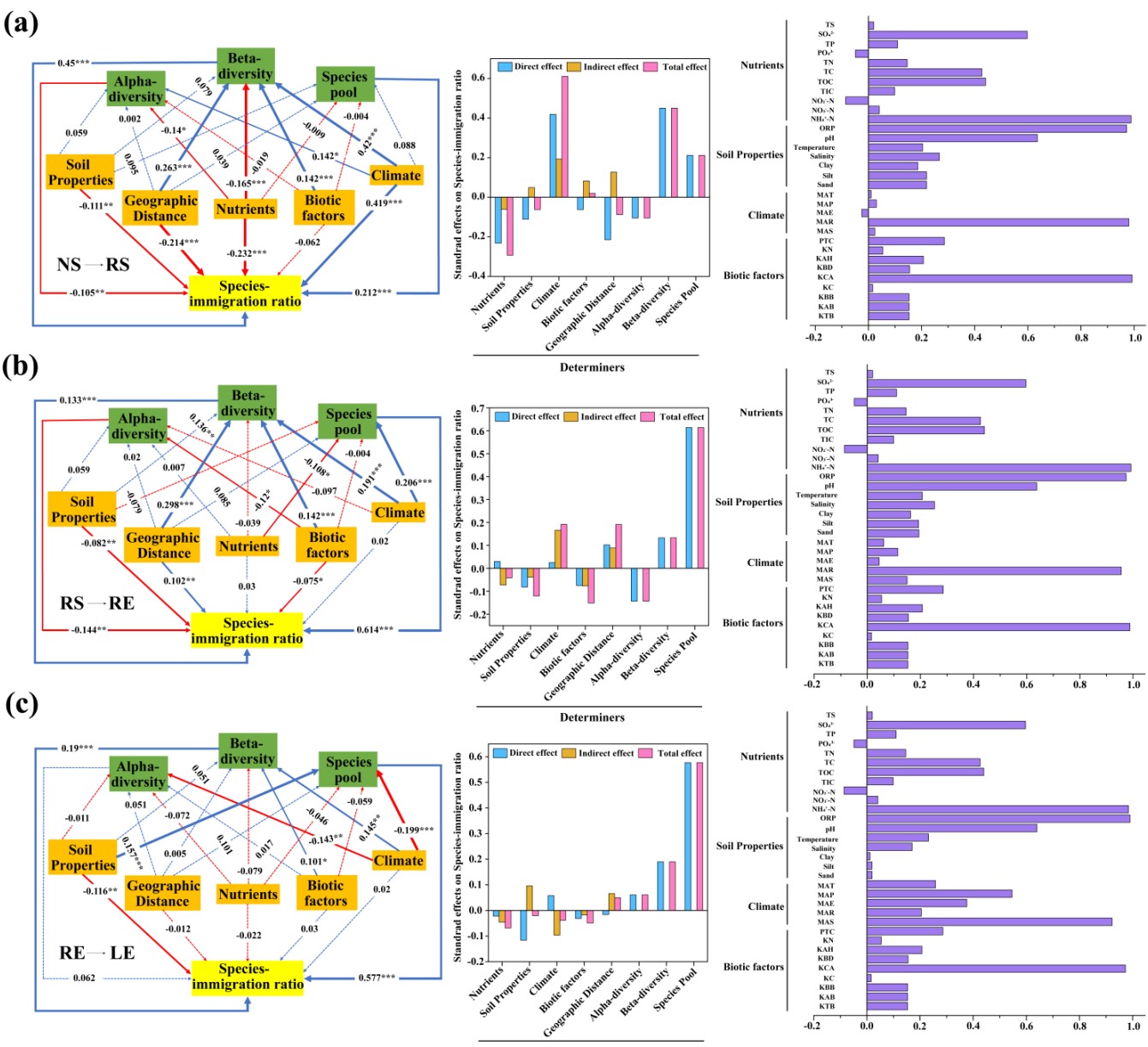

**FIG 7** Panels on the left show the partial least squares path model analysis of the relationship between species-immigration ratio and soil properties, nutrients, climate, biotic factors, alpha diversity, beta diversity, and species pool. (a) The transmission pathway from nonrhizosphere soil to rhizosphere soil (NS→RS) group (goodness of fit = 0.19); (b) The transmission pathway from rhizosphere soil to root endosphere (RS→RE) group (goodness of fit = 0.19); (c) The transmission pathway from root endosphere to leaves (RE→LE) group (goodness of fit = 0.16). Solid and dashed arrows represent significant and insignificant at *P* < 0.05, respectively. Blue and red arrows indicate positive and negative effects, respectively. Significance levels, *P* < 0.05, **P* < 0.01, and ***P* < 0.001, are indicated using different widths of the solid line arrows, as well as the numbers are path coefficients. The center panels show the direct, indirect, and total effects of different factors on the species-immigration ratio. Panels on the right show the individual factor's contribution to the module. ORP, pH, temperature, salinity, and soil texture (clay, silt, and sand) were summarized as soil properties. TS, $SO_4^{2-}$, TP, $PO_4^{3-}$, TN, TC, TOC, TIC, $NO_2$-N, $NO_3^-$-N, and $NH_4^+$-N were summarized as nutrients. MAT, MAP, MAE, MAR, and MAS were summarized as climate factors. PTC, KN, KAH, KBD, KCA, KC, KBB, KAB, and KTB were summarized as biotic factors. KN. *Kandelia obovata* numbers; KAH. *Kandelia obovata* average height; KBD. *Kandelia obovata* basal diameter; KCA. *Kandelia obovata* canopy area; KC, *Kandelia obovata* coverage; KAB, *Kandelia obovata* aboveground biomass; KBB, *Kandelia obovata* belowground biomass; and KTB, *Kandelia obovata* total biomass.

with a negative impact on the SImR. Furthermore, we observed a gradual decrease in the impact of key factors on the SImR of diazotrophic communities, extending from the soil to the leaves across large-scale mangrove habitats.

## DISCUSSION

There is increasing evidence that the immigration process of the microbial community within the plant-soil system is a widespread phenomenon (1, 2) that plays an important role in the functional microbiome in ecosystem services. Plant-rhizosphere soil hosts diazotrophic communities with high heterogeneity, and environmental filtering effects can drive dissimilarities in diazotrophic community diversity. Variations in biodiversity are regulated by environmental factors and alter the dynamics of diazotrophic communities, thereby mediating the balance between species immigration and extinction (54). However, complex interactions across large-scale mangrove habitats remain unclear. The aim of our study was to explore the effects of biotic and abiotic factors on diazotrophic community diversity, assembly processes, and species immigration mechanisms from below- to aboveground across large-scale mangrove habitats.

In the present study, we found that the community composition and distribution patterns of diazotrophs presented distinct geographic disparities (Fig. 1c), which were affected by environmental selection and dispersal in different mangrove habitats (1, 3, 4). *Bradyrhizobium japonicum, Marinobacterium lutimaris,* and *Agrobacterium tumefaciens* were the most abundant species, indicating their broad ecological niche in interspecies competition (Fig. S1). *Bradyrhizobium* and *Agrobacterium*, the most dominant diazotrophs, commonly exist in rhizosphere soils and roots (55, 56). Numerous *Bradyrhizobium* strains have exhibited remarkable resilience in tolerating alternating cycles of dry and wet conditions as well as anoxic conditions during seasonal transitions, playing a key role in the nitrogen cycle in the mangrove ecosystem (57). In addition, *Marinobacterium* can degrade various hydrocarbons as their sole carbon and energy source and has been found in oil-contaminated areas, including mangrove habitats (36). These results indicate that diazotrophic taxa are ubiquitous across various nutrient environments, highlighting their crucial role as core species for nitrogen fixation in mangrove habitats. Our results also showed a common phenomenon in natural ecosystems, where alpha diversity was higher in soil than in plant leaves (18, 33) (Fig. 2a and b), and the diazotrophic species pool exhibited a vertical distribution pattern from leaves to nonrhizosphere soil (Fig. 2c and d), where the diazotrophic species pool peaked in nonrhizosphere soil and declined toward the leaves (18, 58). The results of the NMDS and ANOSIM analyses showed that the structures of diazotrophic communities in soil samples exhibited greater divergence than those of the plant leaves (Fig. S2), indicating that convergence in community structure may be affected by dynamic community shifting (45). The species pool in the nonrhizospheric soil had a steeper slope with increasing geographical distance (Fig. S3c), indicating that the species pool in the soil underwent a greater change than that of endophyte-rhodophytes at a large geographic scale in mangrove ecosystems (48). These findings further support the idea that a larger species pool causally increases the occurrence of immigration movement in diazotrophic communities, which is driven by a neutral equilibrium between species immigration and extinction (59). Ecologists have long recognized that the community composition and local species pool are influenced by biogeographic and evolutionary processes, which are influenced by the spatial (beta diversity) and temporal expansion of habitats (60). The diversity within (alpha diversity) and among (beta diversity) samples is expressed in the same units of species richness (species pool), thus allowing direct comparison of alpha and beta diversity (61). A larger species pool generally results in higher alpha diversity, while greater heterogeneity in the species pool can lead to higher beta diversity due to varying species selection across sites based on local conditions and interactions (62, 63). The β-deviation of the diazotrophic community across colonization regions or habitats may be strongly influenced by the degree of environmental conditions (48, 64). In our study, the variability in the diazotrophic community structure was strongly correlated with MAT, salinity, PTC, and TOC. Wherein MAT ($R$ = 0.318–0.432, $P$ < 0.001) was confirmed to be the most crucial driving factor (Fig. 3). Zhang et al. (58) also discovered that environmental conditions, such as total TOC and soil pH, emerged as the main driving factors in capturing the intricate patterns of beta diversity among bacterial communities

along the latitudinal gradient of Chinese forests. In addition, the abundance of the *nifH* gene can be limited by carbon availability, leading to structural variations within the diazotrophic community (64). Variations in climate (65), soil type and properties (66), and other aspects of the spatial (local, regional, and global) and temporal scales (fast and slow) (67) might have caused differences in the structure of the diazotrophic community and generated distinct geographic distribution patterns in mangrove ecosystems (23).

Over the past few decades, microbial community assembly has been studied for patterns in species abundance across spatial and temporal scales, i.e., deterministic and stochastic ecological processes (68–70). The abiotic factors (i.e., pH and TS) are more important for driving community assembly than biotic factors (Table S2; Fig. S6). However, the results of VPA confirmed that a major fraction of the variation (84%–90%) in the diazotrophic community was unexplained by biotic and abiotic factors. In our study, we found that diazotrophic community assembly was mainly controlled by stochastic processes (Fig. S5). It is well known that dispersal limitation can shape microbial community composition across both habitats and localities within the root system of plants, allowing for ecological drift of diazotrophic communities within large-scale mangrove habitats (59, 71). Stochastic birth and death events, coupled with constrained cellular mobility within diazotrophs in rhizosphere soil or plant tissues over ecological time scales, engender the assembly of community composition because of parent and offspring proximity (71). The differences in the assembly of diazotrophic communities from below to aboveground may be governed by geographic distance, root-derived exudates, and leaf traits (71–73). Plant functional characteristics may regulate the features of their rhizosphere soil to promote microbiome growth, which improves plant fitness in natural mangrove ecosystems (74), thereby leading to stochastic processes (dispersal limitation) dominating the assembly of diazotrophic communities (48, 75). Generally, a balance between deterministic and stochastic processes results in the coexistence of microbial communities (76).

Mangrove roots host a diverse assembly of diazotrophic taxa that play pivotal roles in biological nitrogen fixation in mangrove ecosystems (33, 77). The RACD of the diazotrophic community was calculated to explain the discrepancies in taxon associations by different components (Fig. S7). In our study, the differences in RACD suggested closer associations in the diazotrophic community of the soil and root endosphere than in the leaves (45). Nitrogen-fixing bacteria inhabiting diverse components are positively selected by host microbes through internal and external pathways (52), providing physiological plasticity in nonrhizosphere and rhizosphere soil and thriving within the intricate endophyte-rhizophyte system, with functionally important roles (78). In the present study, *Bradyrhizobium* were abundant in the root endosphere and leaf samples (Fig. 4e); this observation may be attributed to the immigration of diazotrophic bacteria within the soil-plant system. Nitrogen fixation occurs in local communities via diazotrophic taxa from below- to aboveground through internal and external pathways (52). The MacArthur and Wilson theory of island biogeography proposes that species richness in a community is determined by a dynamic balance between immigration and extinction processes (54), which are in turn influenced by the size of the local community's area and its connectivity through dispersal (79). The interplay between spatial and temporal scales has reshaped the landscape of microbial diversity, which emerges at any scale as a result of diverse mechanisms spanning from local to regional levels (80). Our study found a gradually increasing trend for SERr and SImR in diazotrophic communities from the soil to leaves and a gradually decreasing trend for SExR (Fig. 5). In fact, a previous study discovered that despite being relatively stable in species, large changes in species composition occur over time (81). These findings represent ecological trade-offs that arise within species undergoing habitat changes in pursuit of specific nutritional requirements and dominance in niches characterized by microscale abiotic heterogeneity (80).

Biodiversity encompasses not only species richness, quantified as the number of species, but also the elements of species identity, dominance, and rarity (46).

Consequently, even if local species extinction is balanced by species immigration, these species extinctions are not random in terms of species identity and functional performance, and changes in species composition are likely to exert significant ramifications on ecosystem functioning (46). Hence, biodiversity change can be conceptualized as a process of dynamic balance; that is, via the intricate internal framework of endophyte-rhizophyte systems, species exchange occurs to maintain a balance between the species extinction ratio and species immigration ratio (45). In our analysis, we observed that SERr, SExR, and SImR increased as geographic distance increased and were simultaneously significantly correlated with biodiversity (alpha-diversity, beta-diversity, and species pool) (Fig. S11 and S12). Notably, SERr, SExR, and SImR exhibited extremely positive correlations ($P < 0.001$) with beta diversity (Fig. S12b, e, and h). The interplay between habitat filtering and biotic interactions shapes local communities, leading to local extinctions and a reduction in the local species pool (82). These results indicate that the processes of species exchange, extinction, and immigration play crucial roles in maintaining and shaping biodiversity within ecosystems in geographical space. In addition, studying the effects of biotic and abiotic factors on the immigration process of diazotrophs is essential to reveal the mechanisms of biodiversity turnover (46). In our study, we observed that abiotic factors, such as soil properties, nutrients, and climate, were significantly correlated with species exchange, extinction, and immigration of diazotrophic bacteria in mangrove soil. As mentioned above, soil hosts diazotrophic communities, and the immigration process is controlled by geographical distance, root exudates, and plant traits (71–73). Nevertheless, biotic factors, especially biomass, determine the migration of diazotrophic bacteria within plant tissues and can be attributed to the stable nutritional status and more conducive conditions for plant microbiome growth exhibited by mangrove plants. Species immigration not only alters local biodiversity and community dynamics but also exerts a significant effect on nitrogen fixation efficiency and the nitrogen cycling process within mangrove ecosystems (48, 83).

The root-associated soil of mangrove systems constitutes a crucial reservoir of nutrients that are essential for plant growth (18), yet environmental filtering has a significant impact on the structure of microbial communities (48). PLS-PM analysis showed that the species pool was the most important factor directly affecting the SImR of diazotrophic communities, leading to increased SImR (Fig. 7a through c), and soil properties had a direct negative effect on SImR for diazotrophic communities (Fig. 7a). Nevertheless, it is noteworthy that alpha diversity exerted a direct detrimental effect on the SImR of diazotrophic communities in mangrove roots. This observation aligns with the linear regression relationship, in which the SImR of diazotrophic communities decreased with increasing alpha diversity (Fig. S12g). These results demonstrate that the SImR of diazotrophic communities is limited by the species pool in large-scale mangrove habitats and is regulated by soil properties (ORP and pH) (48, 75). In conclusion, this study emphasizes the potential driving factors of the immigration mechanisms of diazotrophic communities, thereby achieving a dynamic balance between species extinction and migration across large-scale mangrove habitats.

## Conclusion

Here, we have demonstrated that changes in environmental pressure on microbial community structure cause patterns of ecological trade-offs in diazotrophic communities across large-scale mangrove habitats. In our study, alpha and beta diversity exhibited significant divergence at different sites and components. Furthermore, the species pool showed a vertical distribution pattern that gradually increased from the leaves to the nonrhizosphere soil at each site. *Bradyrhizobium japonicum, Marinobacterium lutimaris,* and *Agrobacterium tumefaciens* were the main nitrogen-fixing bacterial species that performed a series of nitrogen fixation activities. These species share and migrate among different components through internal and external pathways over a broad mangrove geographical scale. Furthermore, stochastic processes dominate the assembly

of diazotrophic communities in plant-soil systems in large-scale mangrove ecosystems. Climate factors, particularly MAT, can significantly influence the structure of diazotrophic communities. PLS-PM analysis revealed that the species-immigration ratio of diazotrophic communities was mainly limited by the species pool rather than beta diversity and regulated by soil properties, particularly ORP and pH, in mangrove ecosystems. Overall, our findings promote the understanding of extensive species immigration of diazotrophic communities in the context of environmental disturbance and climate change within mangrove habitats and will help enhance the knowledge of colonization routes and dispersal modes of microbial communities in southern China.

## ACKNOWLEDGMENTS

We would like to thank Miss Qun Xie at Analytical and Testing Center of Guangdong Ocean University for their assistance with nutrient analysis. Additional technical support for field material was equally provided by Guangdong Provincial Observation and Research Station for Tropical Ocean Environment in Western Coastal Waters (GSTOEW) and Construction of Key Disciplines in Ocean Science at High-level Universities (231420003 and 080503032101).

This research was financially supported by the National Natural Science Foundation of China (No. 41966005); Guangdong University Innovation Team (Early-warning of marine disasters) (No. 2023KCXTD015) and Scientific Research Start Funds of Guangdong Ocean University; the Open Research Fund Program of Guangxi Key Lab of Mangrove Conservation and Utilization (GKLMC-202001); and the Foundation of Guangxi Academy of Sciences under contract (No. 2023GMRC-02).

N.L., L.P., and N.Li had the initial idea, which was discussed with all authors. H.Z. and L.P. originally designed the long-term experiments. N.L., S.Y., X.Q., J.H., and X.L. planned the methodology. L.P., H.Z., and X.S. collected the data. N.L., H.Z., K.D., Q.H., Q.C., P.W., G.J., and N.Li wrote the R-script for data analysis. N.L. processed the samples, analyzed the data, and wrote the manuscript, to which L.P. and N.L. provided key contributions. All authors commented on previous drafts and approved the final version for publication.

All authors have read and approved the manuscript being submitted and agreed to its submission to this journal. The authors declare that they have no known competing financial interests or personal relationships that could have appeared to influence the work reported in this paper.

## AUTHOR AFFILIATIONS

[1]Laboratory for Coastal Ocean Variation and Disaster Prediction, College of Ocean and Meteorology, Guangdong Ocean University, Zhanjiang, China

[2]Key Laboratory of Climate, Resources and Environment in Continental Shelf Sea and Deep Sea of Department of Education of Guangdong Province, Guangdong Ocean University, Zhanjiang, China

[3]College of Environmental Science and Engineering, Guilin University of Technology, Guilin, China

[4]Guangxi Key Lab of Mangrove Conservation and Utilization, Guangxi Academy of Marine Sciences (Guangxi Mangrove Research Center), Guangxi Academy of Sciences, Beihai, China

[5]Key Laboratory of Environment Change and Resources Use in Beibu Gulf, Ministry of Education (Nanning Normal University), Nanning, China

[6]School of Agriculture, Ludong University, Yantai, China

[7]Department of Biological Sciences, Kyonggi University, Suwon-si, Gyeonggi-do, South Korea

[8]Key Laboratory of the Coastal and Wetland Ecosystems (Xiamen University), Ministry of Education, College of the Environment and Ecology, Xiamen University, Xiamen, China

[9]Key Laboratory of Marine Ecosystem Dynamics, Second Institute of Oceanography, Ministry of Natural Resources, Hangzhou, China

## AUTHOR ORCIDs

Huaxian Zhao http://orcid.org/0000-0002-5997-7671
Nan Li http://orcid.org/0000-0003-1782-6610

## FUNDING

| Funder | Grant(s) | Author(s) |
|---|---|---|
| MOST \| National Natural Science Foundation of China (NSFC) | No. 41966005 | Nan Li |
| Guangdong University Innovation Team (Early-warning of marine disasters)) | 2023KCXTD015 | Nan Li |
| the Open Research Fund Program of Guangxi Key Lab of Mangrove Conservation and Utilization | GKLMC-202001 | Huaxian Zhao |
| the Foundaion of Guangxi Academy of Sciences under contract | No. 2023GMRC-02 | Lianghao Pan |

## DATA AVAILABILITY

The original sequence data were deposited in GenBank under BioProject Accession PRJNA770021 and PRJNA1130613.

## ADDITIONAL FILES

The following material is available online.

### Supplemental Material

**Supplemental Material (mSystems00307-24-s0001.docx).** Supplemental materials and methods, figures, and tables.

### Open Peer Review

**PEER REVIEW HISTORY (review-history.pdf).** An accounting of the reviewer comments and feedback.

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
