## [Reviewer comments · mSystems]

Species pool and soil properties in mangrove habitats influence the species-immigration process of diazotrophic communities across southern China

Nengjian Liao, Lianghao Pan, Huaxian Zhao, Shu Yang, xinyi Qin, Jiongqing Huang, Xiaoli Li, Ke Dong, Xiaofang Shi, Qinghua Hou, Qingxiang Chen, Pengbin Wang, Gonglingxia Jiang, and Nan Li

Corresponding Author(s): Nan Li, Guangdong Ocean University

Review Timeline:

Submission Date:	March 18, 2024
Editorial Decision:	April 30, 2024
Revision Received:	June 6, 2024
Accepted:	June 15, 2024

Editor: Haiyan Chu

Reviewer(s): Disclosure of reviewer identity is with reference to reviewer comments included in decision letter(s). The following individuals involved in review of your submission have agreed to reveal their identity: Yong Li (Reviewer #1); Jun Yuan (Reviewer #2)

Transaction Report:

DOI: <https://doi.org/10.1128/mSystems.00307-24>

Re: mSystems00307-24 (Species pool and soil properties in mangrove habitats influence the species-immigration process of diazotrophic communities across southern China)

Dear Prof. Nan Li:

Revision Guidelines

Sincerely,
Haiyan Chu
Editor
mSystems

Reviewer #1 (Comments for the Author):

This study explored the immigration of diazotrophic communities from soil to leaves across six natural mangrove habitats in China, focusing on the impact of biotic and abiotic factors on functional microbial communities. Using a null model, the author analyzed the assembly processes of diazotrophic communities in various regions. Additionally, the research highlighted the significant role of the species pool and soil properties, such as MAT and pH, in influencing species migration. These findings

enhance our understanding of species dispersal dynamics between belowground and aboveground habitats within mangrove ecosystems. Although the study is data-rich and well-written, it lacks adequate detail concerning the categorization of research subjects and the explanation of species migration mechanisms. Moreover, the presentation and verification of results require careful formatting and significance analysis. Comments for revision are provided below.

1. Why is there a division of regions into different latitudes? The latitude span here is relatively small (6 degrees), while the longitude has a larger variation (10 degrees).
2. Lines 459-461: Is the species pool related to alpha and beta diversity? This relationship lacks detailed explanation and in-depth discussion about how alpha and beta diversity influence species pools.
3. The significance markers in Fig. S9, such as 'a', 'd', 'f', 'g', 'h', appear confusing. Please review and clarify these results.
4. Lines 70-102: The introduction discussing microorganisms with BNF function should be more concise as it does not seem to effectively introduce the topic.
5. Please adjust the orientation of the vertical coordinate in Figure 3, for instance, tilting it to 45 degrees to avoid confusion.
6. In Figure 4a, are there significant differences in the number of ASVs at different locations? Consider removing the error bars unless significance analysis results are provided.
7. Figure 5: Please include significance analysis results.
8. Lines 57-63: This text is overly lengthy. Please consider shortening it.
9. Lines 67-69: This statement does not adequately express a cause and effect relationship. Please revise for clearer expression.
10. Line 123: Replace "no gap is mentioned here" with "Here."
11. Lines 266-268: Please provide specific statistical data.
12. Lines 351 and 532: Remove the word "all."
13. Line 434: The amplicon sequencing does not characterize active microorganisms, please rephrase it.
14. Lines 461-464: Consider splitting the content after "and MAT" into a separate sentence for clarity.
15. Line 475: There is a syntax error. Please rewrite this text for clarity.
16. Figure S8 is crucial for interpreting the results and should be included as a main figure, displayed together with Figure 4.

Reviewer #2 (Comments for the Author):

This article primarily focuses on the assembly and migration of diazotrophic communities within mangrove habitats, elucidating the impacts of both biotic and abiotic factors on diazotrophic microbial populations. Several issues requiring clarification or improvement are identified within the manuscript:

The ecological significance or potential applications of understanding the migration of diazotrophic communities are not adequately addressed in the manuscript.

Lines 36-38: Can "species-immigration mechanisms" be regulated, besides being investigated and discovered?

Lines 130-132: This study only addresses diazotrophic communities and could not reflect the migration status of the entire microbial community.

Lines 150-165: A more detailed description of the methods employed for detecting soil biochemical factor is necessary.

Line 165: Parentheses are colored red; please amend to black.

Lines 205-206: Which function was used to calculate the "species pool"?

Lines 206-208: Reference to the R package is missing.

Lines 210-211: Further elaboration is needed on the method used to assess community assembly.

Lines 211-212: Can ggplot2 be directly used for conducting linear regression analysis and obtaining relevant results, in addition to being used for plotting linear regression graphs?

Lines 212-213: An explanation is required on how Groups I-IV were classified.

Lines 255-256: Why does biodiversity increase with increasing geographical distance? What factors drive this?

Lines 270-272: Which of the biotic and abiotic factors is more important in driving the assembly of diazotrophic communities?

Lines 418-421: The font does not match the rest of the text.

The bar plot in Figure 7 lacks a title for the x-axis.

Dear editor,

Thank you very much for your kind letter on April 30, 2024 concerning our manuscript (mSystems00307-24) and for inviting us to submit a revised version of the manuscript.

We have revised the manuscript in accordance with the reviewers' comments and suggestions, and carefully proof-read the manuscript. Please see below for our point-to-point responses to the reviewers' comments and suggestions.

We would like to thank all the reviewers for useful advice, critical comments and valuable suggestions. We greatly appreciate editor's time and excellent guidance to help us to make our manuscript publishable.

If you have any questions, please don't hesitate to contact me.

I look forward to hearing from you soon.

With the best kindly regards;

Nanli

Prof. Nan Li

Key Laboratory of Climate, Resources and Environment in Continental Shelf Sea and Deep Sea of Department of Education of Guangdong Province, Department of Oceanography, Key Laboratory for Coastal Ocean Variation and Disaster Prediction, College of Ocean and Meteorology, Guangdong Ocean University, Zhanjiang 524088, People's Republic of China.

E-mail: nli0417@163.com

Reviewer: 1

Comments to the Author

Reviewer Comments:

This study explored the immigration of diazotrophic communities from soil to leaves
across six natural mangrove habitats in China, focusing on the impact of biotic and
abiotic factors on functional microbial communities. Using a null model, the author
analyzed the assembly processes of diazotrophic communities in various regions.
Additionally, the research highlighted the significant role of the species pool and soil
properties, such as MAT and pH, in influencing species migration. These findings
enhance our understanding of species dispersal dynamics between belowground and
aboveground habitats within mangrove ecosystems. Although the study is data-rich
and well-written, it lacks adequate detail concerning the categorization of research
subjects and the explanation of species migration mechanisms. Moreover, the
presentation and verification of results require careful formatting and significance
analysis. Comments for revision are provided below:

1. Why is there a division of regions into different latitudes? The latitude span here is
relatively small (6 degrees), while the longitude has a larger variation (10 degrees).

**Response :** Many thanks. Mangrove forests in China are mainly distributed along the
southern and southeastern coastal regions, including provinces such as Guangdong,
Guangxi, Fujian, and Hainan (1). Generally, an increase in temperature and
precipitation is expected in regions with lower latitudes, whereas a decrease in these
parameters is expected in northward transplantation (climate cooling) at northern
latitudes along the coastal areas (2). Climate is an important environmental
determinant of species distribution, and climate change has significant impacts on
biodiversity, including species distribution and interspecific interactions (3). In this
study, we utilized longitude and latitude as geographical distances to investigate their
effects on biodiversity. We feel to mark the samples based on latitude to highlight the

differences in the distribution of diazotrophic communities between northern-southern
regions of mangrove forests, facilitating a better comprehension for readers. However,
if you believe in using longitude to mark the samples is more appropriate, we can
make that adjustment.

The above reference mentioned had been listed as the following:

- 1. Hu W, Wang Y, Zhang D, Yu W, Chen G, Xie T, Liu Z, Ma Z, Du J, Chao B, Lei
G. 2020. Mapping the potential of mangrove forest restoration based on species
distribution models: A case study in China. *Sci Total Environ* 748: 142321.
<https://doi.org/10.1016/j.scitotenv.2020.142321>
- 2. Liang Y, Jiang Y, Wang F, Wen C, Deng Y, Xue K, Qin Y, Yang Y, Wu L, Zhou J,
Sun B. 2015. Long-term soil transplant simulating climate change with latitude
significantly alters microbial temporal turnover. *ISME J* 9:2561-2572.
<https://doi.org/10.1038/ismej.2015.78>
- 3. Zheng J, Wei H, Chen R, Liu J, Wang L, Gu W. 2023. Invasive trends of *Spartina*
*Alterniflora* in the southeastern Coast of China and potential distributional
impacts on mangrove forests. *Plants* 12:1923.
<https://doi.org/10.3390/plants12101923>

2. Lines 459-461: Is the species pool related to alpha and beta diversity? This
relationship lacks detailed explanation and in-depth discussion about how alpha and
beta diversity influence species pools.

**Response :** Thank you very much for your valuable comments. To explore the
relationships between species pool and diversity (alpha- and beta-diversity), linear
regression analysis shown in Figure S4. Corresponding results and discussion section
as the flowing:

“In addition, linear regression analysis showed a positive correlation ($P < 0.001$)
between species pool and diversity (alpha diversity and beta diversity) (Fig. S4),
indicating resource availability, interspecific relationships, and habitat differences
have significant influences on species pool (53).” (Clean Version line: L263-266)

“Ecologists have long recognized that the community composition and local species
 pool are influenced by biogeographic and evolutionary processes, which are
 influenced by the spatial (beta diversity) and temporal expansion of habitats (60). The
 diversity within (alpha diversity) and among (beta diversity) samples is expressed in
 the same units of species richness (species pool), thus allowing direct comparison of
 alpha and beta diversity (61). A larger species pool generally results in higher alpha
 diversity, while greater heterogeneity in the species pool can lead to higher beta
 diversity due to varying species selection across sites based on local conditions and
 interactions (62, 63).” (Clean Version line: L472-480)

 **Fig. S4 Linear regressions for alpha- (a) and beta-diversity (b) associated with**
 **species pool of the diazotrophic communities.** NS: nonrhizosphere sample; RS:
 rhizosphere sample; RE: root endosphere sample; LE: leaves sample.

 3. The significance markers in Fig. S9, such as 'a', 'd', 'f', 'g', 'h', appear confusing.
 Please review and clarify these results.

**Response :** Thank you very much for your good comments and numerous valuable
 suggestions. We have revised the legend of Fig. S10 (previous Fig. S9) to explain the
 meaning of the different letters (significant difference at $P < 0.05$.) and made
 corrections to Fig. S10 to avoid confusion.

**Fig. S10 The variation in immigration mechanisms of diazotrophic communities**

**by boxplot.** The values of species-exchange ratio (SERr) (1), (4), (7),

species-extinction ratio (SExR) (2), (5), (8), and species-immigration ratio (SImR) (3),

(6), (9) in different mangrove sites along latitude gradient. Different letters above the

box plot on the indicated a significant difference ($P < 0.05$) using the Wilcoxon test.

NS→RS, the transmission pathway from nonrhizosphere soil to rhizosphere soil;

RS→RE, the transmission pathway from rhizosphere soil to root endosphere;

RE→LE, the transmission pathway from root endosphere to leaves. LD: Ledong

mangrove site; DZ: Dongzhaigang mangrove site; SK: Shankou mangrove site; ZZ:

Zhenzhuwan mangrove site; HD: Huidong mangrove site; FG: Fugong mangrove site.

NS: nonrhizosphere sample; RS: rhizosphere sample; RE: root endosphere sample;

LE: leaves sample.

4. Lines 70-102: The introduction discussing microorganisms with BNF function

should be more concise as it does not seem to effectively introduce the topic.

**Response** : Thank you very much for your important comments and valuable
suggestions. Your advice helps us significantly improve the manuscript quality. We
have revised the sentences as the following described:

“Diazotrophic communities are essential components of the nitrogen cycle of
terrestrial ecosystems, transforming substantial amounts of atmospheric N₂ into
available N via biological nitrogen fixation (BNF) (20, 21). Molecular methods based
on universal PCR detection of *nifH* marker genes have been successfully applied to
detect diazotroph communities in the natural environment (22-24). Compared to
free-living N fixation (FLNF)diazotrophs, symbiotic diazotrophs may have an
advantage because they live within plant tissues, where better niches are established
for N₂ fixation and assimilation (25, 26). Recent evolutionary origins and ecological
studies of nitrogen fixation (*nif*) genes and their genomic information (27, 28) indicate
that *nif*-carrying free-living members derived from diverse soil samples have
independently evolved from symbiotic ancestors. Furthermore, recent studies have
shown that cooccurrence of endophytic diazotrophs is widespread in roots, stems, and
leaves (29-31). These studies suggest that plant roots harbour diverse symbiotic
diazotrophs that are genetically adapted to a dynamic environment. They can be
naturally transferred through the soil, spread systemically, and reach aerial
components. However, the transmission processes and environmental selection of
rhizospheric and endophytic diazotrophs in natural ecosystems, particularly
subtropical forest ecosystems, remain poorly understood.” (Clean Version line:
L90-107)

5. Please adjust the orientation of the vertical coordinate in Figure 3, for instance,
tilting it to 45 degrees to avoid confusion.

**Response :** According to the suggestion, we have made corrections to the graph.

6. In Figure 4a, are there significant differences in the number of ASVs at different
locations? Consider removing the error bars unless significance analysis results are
provided.

**Response :** Thank you very much for your good suggestion. According to the
comments, we have removed the error bars. please check the new Figure 4.

7. Figure 5: Please include significance analysis results.

**Response :** Thank you very much for your important comments. According to the
 comments, we have added the statistical results in the new Figure 5. We also revised
 the legend of Figure 5 to explain the meaning of the different letters.

**Fig. 5.** Variations of the immigration process of diazotrophic communities

(species-exchange ratio (SErR), species-extinction ratio (SExR), and
species-immigration ratio (SIrR) in different components across mangrove
habitats. Different letters above the box plot indicated a significant difference
($P < 0.05$) using the Wilcoxon test. NS→RS, the transmission pathway from
nonrhizosphere soil to rhizosphere soil; RS→RE, the transmission pathway from
rhizosphere soil to root endosphere; RE→LE, the transmission pathway from root
endosphere to leaves. NS: nonrhizosphere sample; RS: rhizosphere sample; RE: root
endosphere sample; LE: leaves sample.

8. Lines 57-63: This text is overly lengthy. Please consider shortening it.

**Response :** Many thanks for your good comments and numerous valuable suggestions.

We have revised the sentences as the following described:

“To explore transmission processes at both the individual plant and ecosystem levels,
we need to study endophyte colonization routes and dispersal modes, for example,
microbial communities assemble and immigration mechanisms between belowground
(i.e., soil and root) and aboveground (i.e., leaf) host environment.” (Clean Version line:

L79-82)

9. Lines 67-69: This statement does not adequately express a cause and effect
relationship. Please revise for clearer expression.

**Response :** Thank you very much for your valuable comments. We have revised the
sentences as the following described:

“Mangrove ecosystems are rich in organic carbon and nitrogen, making them ideal
habitats for microbial communities, it is necessary to evaluate the migration or

transmission processes between below- and aboveground habitats in the functional
microbiome.” (Clean Version line: L86-89)

10. Line 123: Replace "no gap is mentioned here" with "Here."

**Response :** Many thanks. According to the comments, we have revised the sentences
as the following described:

“Here, we collected sediment, rhizosphere soil, root, and leaf samples from the
popular species *Kandelia obovata* from coastal mangroves across six natural coastal
mangrove wetlands in China, spanning 1,150 km with a latitudinal range from
18.44°N to 24.39°N and a longitudinal range from 108.24°E to 117.92°E.” (Clean
Version line:128-131)

11. Lines 266-268: Please provide specific statistical data.

**Response :** Thank you very much for your important comments and valuable
suggestions. We have added specific statistical results in the sentences as the flowing
described:

“Soil texture, including sand, silt, and clay ($R = 0.217-0.426$, $P < 0.001$), was more
important for the root-associated diazotrophic community than for the leaves.
However, SO_4^{2-} ($R = 0.322$, $P < 0.01$) was the main driving factor affecting the
diazotrophic community in the rhizosphere soil.” (Clean Version line:273-276)

12. Lines 351 and 532: Remove the word "all."

**Response :** Many thanks. We have revised the manuscript, please check that in the new
version. (Clean Version line:366 and 554)

13. Line 434: The amplicon sequencing does not characterize active microorganisms,
please rephrase it.

**Response :** Thank you very much for your good comments and numerous valuable
suggestions. We have revised the sentence as the following described:

“*Bradyrhizobium japonicum*, *Marinobacterium lutimaris*, and *Agrobacterium*
*tumefaciens* were most abundant species, indicating their broad ecological niche in
interspecies competition (Fig. S1).” (Clean Version line:447-449)

14. Lines 461-464: Consider splitting the content after "and MAT" into a separate
sentence for clarity.

**Response :** Many thanks. According to the comments, we have revised the sentence as
the following described:

“In our study, the variability in the diazotrophic community structure was strongly
correlated with MAT, salinity, PTC, and TOC. Wherein MAT ($R = 0.318-0.432$, $P <$
0.001) was confirmed to be the most crucial driving factor (Fig. 3).” (Clean Version
line:482-485)

15. Line 475: There is a syntax error. Please rewrite this text for clarity.

**Response :** Thank you very much for your valuable suggestions. According to the
comments, we have revised the sentences as the following described:

“Over the past few decades, microbial community assembly has been studied for
patterns in species abundance across spatial and temporal scales, i.e., deterministic
and stochastic ecological processes. (68-70)” (Clean Version line:495-497)

16. Figure S8 is crucial for interpreting the results and should be included as a main
figure, displayed together with Figure 4.

**Response :** Thank you very much for your valuable comments. We have changed the
previous Figure 4 and Figure S8 to the new Figure 4 and revised the legend of Figure
4. Please check the new graph.

**Fig. 4. Distribution patterns of the shared and unique ASVs in different groups.**

The UpSet plot was chosen to visualize the common and unique mean ASVs (a)
 across different components. The two potential immigration pathways of diazotrophic
 ASVs (b) and species (c) in different components from belowground (i.e.,
 nonrhizosphere soil and rhizosphere soil) to aboveground (i.e., root endosphere and
 leaves) habitats were predicted. The shared ASVs and species from distinct
 components were defined as Group I–IV (Group I: the ASVs shared between NS and
 RS, Group II: the ASVs shared between NS, RS, and RE, Group III: the ASVs shared
 between NS, RS, RE, and LE, Group IV: the ASVs shared between NS and LE).
 Groups I, II, and III contained the shared ASVs (b) and species (c) that likely
 immigrated through internal pathway; Group IV contained the shared ASVs (b) and
 species (c) that likely immigrated through external pathway. Distributions of
 diazotrophic composition (top 13) in Groups I–IV at the class level (d) and species

level (e), in which Group I indicated the ASVs shared between NS and RS; Group II
indicated the ASVs shared between NS, RS, and RE; Group III indicated the ASVs
shared between NS, RS, RE, and LE; Group IV indicated the ASVs shared between
NS and LE. NS: nonrhizosphere sample; RS: rhizosphere sample; RE: root
endosphere sample; LE: leaves sample.

Reviewer: 2

Comments to the Author

This article primarily focuses on the assembly and migration of diazotrophic
communities within mangrove habitats, elucidating the impacts of both biotic and
abiotic factors on diazotrophic microbial populations. Several issues requiring
clarification or improvement are identified within the manuscript:

The ecological significance or potential applications of understanding the migration of
diazotrophic communities are not adequately addressed in the manuscript.

1. Lines 36-38: Can "species-immigration mechanisms" be regulated, besides being
investigated and discovered?

**Response :** Species-immigration mechanisms are regulated by biodiversity and
environmental conditions. For instance, habitat destruction, environmental pollution
and anthropogenic global changes (e.g. climate change) pose severe challenges to
local habitat biodiversity (1). Habitat loss has pervasive and disruptive impacts on
biodiversity, causing exceptionally high for the rates of species extinctions (2, 3). In
our study, pearson's correlation analysis and least squares path modelling (PLS-PM)
show that species pool and soil properties can significantly influence the
species-immigration mechanisms. However, we are not available to utilize specific
isotope-labeled compounds (such as ^{13}C , ^{15}N , etc.) to track their metabolism and
transformation processes for diazotrophic microbial communities within the
rhizosphere soil-plant system. In this study, we used the conventional ecological
methods such as species composition, diversity analysis, and community assembly,
combined with the model was described by Hillebrand and Sun et al (3, 4), assessing
the migration mechanisms of diazotrophic microbial communities within the
rhizosphere mangrove soil-plant systems. From an ecological perspective, this study
offers new insights for understanding the ecological trait diversity patterns and spread
pathways of functional microbial communities between below and aboveground
habitats in natural ecosystems.

The above reference mentioned had been listed as the following:

- 1. Rai PK, Singh JS. 2020. Invasive alien plant species: Their impact on
environment, ecosystem services and human health. *Ecol Indic* 111:106020.
<https://doi.org/10.1016/j.ecolind.2019.106020>
- 2. Ewers RM, Didham RK. 2006. Confounding factors in the detection of species
responses to habitat fragmentation. *Biol Rev* 81:117-142.
<https://doi.org/10.1017/S1464793105006949>
- 3. Hillebrand H, Blasius B, Borer ET, Chase JM, Downing JA, Eriksson BK, et al.
2018. Biodiversity change is uncoupled from species richness trends:
Consequences for conservation and monitoring. *J Appl Ecol* 55:169-184.
<https://doi.org/10.1111/1365-2664.12959>
- 4. Sun R, Chen Y, Han W, Dong W, Zhang Y, Hu C, Liu B, Wang F. 2020. Different
contribution of species sorting and exogenous species immigration from manure
to soil fungal diversity and community assemblage under long-term fertilization.
*Soil Biol Biochem* 151:108049. <https://doi.org/10.1016/j.soilbio.2020.108049>

2. Lines 130-132: This study only addresses diazotrophic communities and could not
reflect the migration status of the entire microbial community.

**Response :** Thank you very much for your valuable comments. we have revised the
sentences as the following described:

“This work is crucial to facilitate a better understanding of immigration mechanisms
of diazotrophic communities across mangrove habitats.” (Clean Version line:135-137)

3. Lines 150-165: A more detailed description of the methods employed for detecting
soil biochemical factor is necessary.

**Response :** Thank you very much for your valuable comments. A more detailed
description of the methods employed for detecting soil biochemical factor are
provided in the Supplementary Information on Materials and Methods. As the
following described:

“For soils in the nonrhizosphere, soil properties, including oxidation–reduction

potential (ORP), salinity, temperature, and pH, were measured using a pH and ORP
meter (HI98121; Hanna, Italy). Soil texture (sand, silt, and clay) was determined
using a Malvern Mastersizer 2000 (Malvern, United Kingdom) (Christensen and
Olesen, 1998). Nutrients, including total carbon (TC), total organic carbon (TOC),
total nitrogen (TN), and total sulfur (TS), were determined using an elemental
analyzer (Vario Macro Cube, Germany). The total inorganic carbon (TIC) was defined
as TC minus TOC. Total phosphorus (TP) was measured as previously described by
Wang et al. (1). Soil water extracts were used to measure SO_4^{2-} and PO_4^{3-} via ion
chromatography (ICS-2100; Dionex, USA). Inorganic nitrogen (NO_2^- -N, NO_3^- -N, and
NH_4^+ -N) was extracted using 2 M KCl and determined using a continuous-flow
analyzer (SAN++; Skalar, Breda, the Netherlands). Biotic factors (total plant coverage,
*K. obovata* number, *K. obovata* average height, *K. obovata* basal diameter, *K. obovata*
canopy area, *K. obovata* coverage) were estimated in 5×5 m plots during the field
survey. In addition, *K. obovata* aboveground biomass, *K. obovata* belowground
biomass, and *K. obovata* total biomass were calculated using allometric equations by
Rahman et al. (2). Data on climatic factors, including the mean annual temperature,
mean annual precipitation, mean annual evaporation, mean annual relative humidity,
and mean annual sunshine duration, were collected from the China Meteorological
Data Sharing Service System (<http://data.cma.cn/>) (Table S1).”

4. Line 165: Parentheses are colored red; please amend to black.

**Response :** Many thanks. We have revised the manuscript as the flowing described:

“Data on climatic factors, including the mean annual temperature, mean annual

precipitation, mean annual evaporation, mean annual relative humidity, and mean
annual sunshine duration, were collected from the China Meteorological Data Sharing
Service System (<http://data.cma.cn/>) (Table S1).” (Clean Version line:165-168)

5. Lines 205-206: Which function was used to calculate the "species pool"?

**Response :** Species pool was estimated using the the “specpool” function by the
“vegan” package in R software. We have revised the sentence as the flowing
described:

“The species pool of the diazotrophic community was estimated using the “specpool”
function in “vegan” package (48).” (Clean Version line:206-208)

6. Lines 206-208: Reference to the R package is missing.

**Response :** Thank you very much for your valuable comments. We have revised the
sentence and added references as the flowing described:

“Nonparametric testing (KwWlx) and Wilcoxon test were performed to reveal the
differences using the “EasyStat” and “stats” packages, respectively (49).” (Clean
Version line:208-209)

7. Lines 210-211: Further elaboration is needed on the method used to assess
community assembly.

**Response :** Thank you very much for your valuable comments. A more detailed
method used to assess community assembly provided in the Supplementary
Information on Materials and Methods. As the following described:

“We calculated the β -nearest taxon index (β NTI) to further infer the assembly
processes of diazotrophic communities (11). β NTI > 2 indicates heterogeneous
selection of a deterministic process, β NTI < -2 indicates homogeneous selection of
the deterministic process, whereas β NTI values between -2 and 2 indicates that the
community assembly was dominated by stochastic processes. Then, community
assembly was further distinguished using the modified Raup-Crick index (RC_{Bray})

within a certain group and chemical property ranges with 999 randomizations, which
is able to indicate dispersal limitation and homogeneous dispersal of the stochastic
processes with $RC_{\text{Bray}} > 0.95$ and $|\beta_{\text{NTI}}| < 2$, $RC_{\text{Bray}} < -0.95$ and $|\beta_{\text{NTI}}| < 2$,
respectively. The fraction of pairwise comparisons with $|\beta_{\text{NTI}}| < 2$ and $|RC_{\text{Bray}}| < 0.95$
was used to estimate the influence of the “non-dominant” fraction (i.e., ecological
drift) (11).”

8. Lines 211-212: Can ggplot2 be directly used for conducting linear regression
analysis and obtaining relevant results, in addition to being used for plotting linear
regression graphs?

**Response :** Thank you very much for your good comments and numerous valuable
suggestions. we have revised the sentences as the following described:

“The linear regression analysis was performed using the “ggplot2” and “ggpmisc”
packages.” (Clean Version line:215-216)

9. Lines 212-213: An explanation is required on how Groups I-IV were classified.

**Response :** Thank you very much for your valuable comments. A more detailed how to
classified Groups I-IV was provided in the Supplementary Information on Materials
and Methods. As the following described:

“The shared ASVs among diazotrophic communities from the four distinct
components were defined as Groups I–IV (Group I: ASVs shared between NS and RS;
Group II: ASVs shared between NS, RS, and RE; Group III: ASVs shared between
NS, RS, RE, and LE; Group IV: ASVs shared between NS and LE), as previously
described by Zhang et al. (13)”

10. Lines 255-256: Why does biodiversity increase with increasing geographical
distance? What factors drive this?

**Response :** Many thanks. Linear regression analysis was conducted, we found that
biodiversity significantly increases with increasing geographic distance. The increase

in biodiversity with increasing geographical distance can be attributed to various
factors for example, species dispersion is universally considered important for
biodiversity conservation (1). Habitat heterogeneity (habitat disruption and changes)
in community structure may impact community activity and ecosystem performance
(2). Finally, historical biogeography such as past climate changes, can shape current
patterns of biodiversity by influencing species distributions and evolutionary histories
(3).

The reference mentioned had been listed as the following:

- 1. Trakhtenbrot A, Nathan R, Perry G, Richardson DM. 2005. Distributions. The
importance of long-distance dispersal in biodiversity conservation. *Divers*
*Distrib* 11:173-181. <https://doi.org/10.1111/j.1366-9516.2005.00156.x>
- 2. Franklin RB, Mills AL. 2009. Importance of spatially structured environmental
heterogeneity in controlling microbial community composition at small spatial
scales in an agricultural field. *Soil Biol Biochem* 41:1833-1840.
<https://doi.org/10.1016/j.soilbio.2009.06.003>
- 3. Bissett A, Richardson AE, Baker G, Wakelin S, Thrall PH. 2010. Life history
determines biogeographical patterns of soil bacterial communities over multiple
spatial scales. *Mol Ecol* 19:4315-4327.
<https://doi.org/10.1111/j.1365-294X.2010.04804.x>

11. Lines 270-272: Which of the biotic and abiotic factors is more important in
driving the assembly of diazotrophic communities?

**Response :** Thank you very much for your important comments. To explore which of
the biotic and abiotic factors is more important in driving the assembly of
diazotrophic communities, the Mantel test (Table S2) and variation partitioning
analysis (new Fig. S6) were performed to reveal the relationships between β NTI and
driving factors (biotic and abiotic factors). Table S2 and new Fig. S6 were provided in
the Supplementary Information. Corresponding data analysis process, results, and
discussion section as the flowing:

“Variation partitioning analysis (VPA) was conducted to examine the contribution of
biotic and abiotic factors to the community assembly according to the RDA (51).”
(Clean Version line:213-215)

“Next, we explored the correlations between driving factors and β NTI using Mantel
tests (Table S2). The results showed that pH ($R = -0.044-0.1243$, $P < 0.001$), TS ($R =$
$-0.0728-0.0806$, $P < 0.001$) determined the changes in diazotrophic community
structure, whereas ORP ($R = 0.1074$, $P < 0.001$) were significantly associated with
nonrhizosphere community structure. To classify the importance of the biotic and
abiotic factors in driving the assembly of diazotrophic communities, variation
partitioning analysis (VPA) was performed (Fig. S6). The VPA results showed that
abiotic factors is more important for driving the assembly of diazotrophic
communities, especially in the nonrhizosphere sample group. However, the assembly
of diazotrophic communities was unexplained (84%–90%) regardless across different
components (Fig. S6).” (Clean Version line:288-298)

“The abiotic factors (i.e., pH and TS) is more important for driving community
assembly than biotic factors (Table S2 and Fig. S6). However, the results of VPA
confirmed that a major fraction of the variation (84-90%) in diazotrophic community
was unexplained by biotic and abiotic factors. In our study, we found that
diazotrophic community assembly was mainly controlled by stochastic processes (Fig.
S5).” (Clean Version line:497-502)

12. Lines 418-421: The font does not match the rest of the text. The bar plot in Figure
7 lacks a title for the x-axis.

**Response :** Thank you very much for your valuable comments. According to the
 comments, we have revised the text format and added a title for the x-axis to the bar
 plot of Figure 7. Please check that in the new version.

Re: mSystems00307-24R1 (Species pool and soil properties in mangrove habitats influence the species-immigration process of diazotrophic communities across southern China)

Dear Prof. Nan Li:

Your manuscript has been accepted, and I am forwarding it to the ASM production staff for publication. Your paper will first be checked to make sure all elements meet the technical requirements. ASM staff will contact you if anything needs to be revised before copyediting and production can begin. Otherwise, you will be notified when your proofs are ready to be viewed.

Sincerely,
Haiyan Chu
Editor
mSystems

Reviewer #2 (Comments for the Author):

Thanks for the efforts on the manuscript, there is no more additional comment.